# The Analogy between Tinnitus and Chronic Pain: A Phenomenological Approach

**DOI:** 10.3390/brainsci13081129

**Published:** 2023-07-27

**Authors:** Arnaud J. Norena

**Affiliations:** Laboratoire de Neurosciences Sensorielles et Cognitives, CNRS, Aix-Marseille University, 13003 Marseille, France; arnaud.norena@univ-amu.fr

**Keywords:** tinnitus, chronic pain, phenomenology, lived experience, perception, sensation, perception of time

## Abstract

Tinnitus is an auditory sensation without external acoustic stimulation or significance, which may be lived as an unpleasant experience and impact the subject’s quality of life. Tinnitus loudness, which is generally low, bears no relation to distress. Factors other than psychoacoustic (such as psychological factors) are therefore implicated in the way tinnitus is experienced. The aim of this article is to attempt to understand how tinnitus can, like chronic pain, generate a ‘crisis’ in the process of existence, which may go as far as the collapse of the subject. The main idea put forward in the present article is that tinnitus may be compared to the phenomenon of pain from the point of view of the way it is experienced. Although the analogy between tinnitus and pain has often been made in the literature, it has been limited to a parallel concerning putative physiopathological mechanisms and has never really been explored in depth from the phenomenological point of view. Tinnitus is comparable to pain inasmuch as it is felt, not perceived: it springs up (without intention or exploration), abolishes the distance between the subject and the sensation (there is only a subject and no object), and has nothing to say about the world. Like pain, tinnitus is formless and abnormal and can alter the normal order of the world with maximum intensity. Finally, tinnitus and pain enclose the subject within the limits of the body, which then becomes in excess. Tinnitus may be a source of suffering, which affects not only the body but a person’s very existence and, in particular, its deployment in time. Plans are thus abolished, so time is no longer ‘secreted’, it is enclosed in an eternal present. If the crisis triggered by tinnitus is not resolved, the subject may buckle and collapse (depression) when their resources for resisting are depleted. The path may be long and winding from the moment when tinnitus emerges to when it assaults existence and its eventual integration into a new existential norm where tinnitus is no longer a source of disturbance.

## 1. Introduction

A recent work by the Association Francophone des Equipes Pluridisciplinaires en Acouphénologie (AFREPA) defines tinnitus as follows: “Tinnitus is an auditory sensation without an external sound stimulation or meaning, which can be lived as an unpleasant experience, possibly impacting quality of life” [1] (other definitions has been proposed [2,3]).

We may distinguish two types of tinnitus: so-called objective tinnitus (rare, estimated at 5% of the total prevalence), which is related to an acoustic source coming from the body (vascular issues, myoclonia of the muscles in the middle ear, etc.), and ‘subjective’ tinnitus (95% of tinnitus occurrence), which is not related to an acoustic cause. Tinnitus may result from a wide range of causes, but the number of these causes is unknown. What all types of tinnitus have in common, on the other hand, is that it is the consequence of neuronal activity in the auditory system generated at a peripheral and/or central level [4].

Tinnitus is highly prevalent, on the order of 15 to 20% of the general population according to an English study carried out on a large cohort [5]. Overall, the prevalence of tinnitus increases with age to reach 25% in the 65- to 69-year-old age group. That study makes a distinction between subjects bothered by tinnitus (those who seek clinical assistance) and those who are not. It is interesting to note that only 20% of subjects seek treatment. Therefore, in contrast, almost 80% of tinnitus subjects would appear to have a good tolerance for tinnitus. This apparent dichotomy between well-tolerated and poorly tolerated conceals a reality that is far more complex. At one extreme, poorly tolerated tinnitus can lead to depression or even suicide, whereas the majority of subjects who consider that they have a good tolerance for their tinnitus report that they would nonetheless be relieved if it were possible to eliminate it [6].

If the psycho-acoustic characteristics of tinnitus, i.e., timbre, pitch, loudness, and localisation, contribute to the discomfort felt by subjects, then their contribution is on the other hand limited and insufficient to explain the subjects’ experience of the condition. It seems that the severity of tinnitus is not or is minimally related to its timbre, pitch, or localisation (but see above) [7]. Intuitively, it might be thought that the severity of tinnitus is proportional to its loudness (subjective intensity): the more tinnitus is perceived as being loud, the more bothersome it is. Yet, if certain studies do indeed show a link between these two variables, this link is statistically very insignificant (~20% of the variance explained) [8]. On the other hand, it is possible that the severity of tinnitus depends on the way it interacts (masking and residual inhibition) with acoustic stimulation. Tinnitus that is easily masked by light background noise and/or that is suppressed after exposure to the same background noise (residual inhibition) is certainly better tolerated than tinnitus that is never masked and is always present [9]. This difference highlights an obvious fact: the impact of tinnitus depends, at least partly, on the proportion of the time during which it is actually perceived. No matter whether tinnitus is intermittent or masked by a certain background noise, it will be all the more severe if it is perceived in a continuous manner and without interruption. This aspect of tinnitus (its degree of continuity or chronicity) should be considered in relation to the notion of respite, which I shall discuss later. This notion of respite is exploited notably in methods based on masking tinnitus. The explicit hypothesis formulated by these approaches is that it is less bothersome to hear a sound with a wide bandwidth over which one has control (white noise) than to hear one’s tinnitus with timbre and pitch that may be more intrusive and more unpleasant. A recent study indeed shows that white noise is judged as neutral (neither agreeable nor disagreeable), whereas pure sounds at high frequency (8 kHz) are judged as unpleasant [10]. The effectiveness of this method thus depends on the level of discomfort procured by the signal used to mask tinnitus: at best, the acoustic signal is perfectly ‘painless’, and it is an effective method of treatment, and at worst, the acoustic signal is as bothersome as tinnitus, and this approach cannot be considered as an option.

The explanation of the discomfort associated with the presence of tinnitus should not be sought solely with regard to the psycho-acoustic characteristics of tinnitus, even if they may play a role in its impact. The aim of the present article is to contribute to the reflection aimed at understanding how tinnitus may induce a disagreeable experience, which may, in extreme cases, result in suffering. The treatment of this issue will take inspiration from phenomenology, a branch of philosophy which aims to revert to the subject’s actual experience.

## 2. Critique of Behavioural and Cognitive Psychology

Various models have been proposed that attempt to describe the disagreeable character of tinnitus and its negative impact on the quality of life of subjects. I shall limit myself here to summarising the two main models which, at the present time, have a certain influence on the community of physicians and researchers working on tinnitus.

### 2.1. Behaviorism

In a pioneering and still influential model issued by the behaviourism school, the negative character of tinnitus is seen as the result of classic or Pavlovian conditioning, which attributes a major role to unconscious psychological processes [11]. Firstly, tinnitus, as a conscious sensation, is related to physio-pathological causes that are the source of an ‘aberrant’ neuronal signal in the central nervous system [12]. What is critical from the point of view of the lived experience, is that once the signal linked to tinnitus is generated (regardless of the mechanism), tinnitus may take on a disagreeable character on the condition that it is associated with a concomitant disagreeable event (illness, separation, bereavement, losing one’s job, failure, etc.). Tinnitus then becomes the conditional stimulus of a reaction that includes the activation of the emotional system and the sympathetic system (negative reinforcement). The model suggests that this association is strengthened by a kind of vicious circle, which may result in considerable discomfort: tinnitus is related to an activation of the sympathetic system and an increase in vigilance, which enhance the detection of tinnitus and further augment the stress system, etc.

“*Tinnitus-related neuronal activity acts as a stimulus and the reaction evoked by activation of the sympathetic part of the autonomic nervous system act as reinforcement. Consequently, once the initial association between tinnitus perception and some negative event happening at the same time develops, the reflex loop is rapidly enhanced, as both stimulus and reinforcement are continuously present*.”(Jastreboff, 2007)

The clinical approach proposed by this model consists of facilitating habituation, that is, ‘breaking’ the association between tinnitus and negative reactions and emotions that are associated with it, on the basis of two complementary approaches. Firstly, a psycho-educative process (‘counselling’) aims to inform the patient of the physio- and psychopathological mechanisms underlying tinnitus and of its harmlessness (in the vast majority of cases, tinnitus is not associated with a serious ailment). In particular, the patient is informed of the vicious circle that is thought to develop, which is the cause of enhanced discomfort. This approach is intended to reduce, as much as possible, any uncertainty regarding the phenomenon of tinnitus (we suffer less from something we understand or know than from something that is new and obscure). It often happens that the simple act of reassuring subjects about the benign nature of tinnitus suffices to assist them [13]. In addition to counselling, an acoustic signal that is neutral from the point of view of emotions and attention (usually a broad-band noise) is used. The purpose of the acoustic signal is to make tinnitus more difficult to detect, and thus to reduce its demands on attention and the activation of the sympathetic system. The acoustic signal is a sensorial complement to the counselling aimed at encouraging habituation.

### 2.2. The Cognitive Model

In contrast to the model described above, a more recent model referred to as ‘cognitive’ explains the disagreeable character of tinnitus by highlighting the conscious mechanisms. Tinnitus would thus possess a disagreeable character because of negative automatic thoughts and beliefs associated with it: ‘tinnitus is associated with a serious disease’, ‘I’m going to go on suffering like this all my life’, ‘the tinnitus will get worse and drive me mad’, ‘tinnitus is ruining my life’, and ‘tinnitus prevents me from living’, etc. [14]. Since tinnitus is interpreted as a threat, its presence is thus associated with an activation of the emotional and sympathetic systems, and habituation (when tinnitus is no longer interpreted as a warning signal) does not occur or, in any case, is delayed. In this context, the subjects may use defensive and survival behaviours to cope with tinnitus, which turn out to be inappropriate in the long term, such as avoidance behaviours, which prevent the process of habituation. If subjects take measures to mask their tinnitus (sound therapy techniques and/or seeking relatively noisy environments), or if they engage in more activities in order to forget their tinnitus (or do not pay any attention to it), they never really face up to their tinnitus, and thus, they do not develop any strategy for acceptance, and it will sooner or later return and disturb their lives. This approach is similar to that used for the treatment of phobias: you can only reduce a phobia by being exposed to the object of the phobia, that is to say, by ‘taming’ it and gradually reducing the emotional explosion triggered by the object of the phobia. It may be noted that this model gives a central role to involuntary psychological processes, which have been adapted by natural selection to be triggered automatically in case of an alert and/or a survival situation. The behavioural and cognitive models suggest that this behavioural repertoire fixed by evolution cannot be adapted to the resolution of the issue. From the clinical point of view, the cognitive models favour altering negative thoughts regarding tinnitus, suppressing counterproductive behaviours, and favouring useful behavioural strategies. The following passage summarises the cognitivist model [14]:

“*The psychological understanding of why tinnitus can be a distressing condition posits that it becomes problematic when it acquires an emotive significance through cognitive processes. Principle therapeutic efforts are directed at reducing or removing the cognitive (and behavioral) obstacles to habituation. (…) The model posits that patients’ interpretations of tinnitus and the changes in behavior that result are given a central role in creating and maintaining distress*.”

### 2.3. Some Reflections on Behaviourism and the Cognitive Model

Jastreboff’s model maintains that the discomfort associated with tinnitus is linked to Pavlovian conditioning. Yet, this hypothesis has never been demonstrated (even if one might just possibly recognise that a distressing event might aggravate the unpleasantness of tinnitus when it appears). Furthermore, the cognitive model suggests that the disagreeable character of tinnitus derives above all from negative automatic thoughts, erroneous beliefs, and inappropriate behaviours. The treatment thus consists of informing subjects of the unfounded nature of their beliefs and the counterproductive character of their survival behaviours. It is worth noting that the cognitive model adds a decisive element to the behaviourist model: the notion of conscious processes. Nevertheless, if these processes are conscious, all the same, they do not escape the will inasmuch as automatic negative thoughts and survival behaviours are adaptive vestiges inherited from evolution. The ‘cognitive revolution’ sheds light on mental processes and a certain liberty that goes with them, which also makes it possible to break free of strict biological determinism. Whereas the ‘Pavlovian brain’ is subjected to simple statistical contingency (events A and B occur at the same time), the ‘cognitive brain’ is that of adaptation by learning. The cognitive brain thus disposes of a certain potential or ‘cognitive reserve’ that is developed and used for the personal accomplishment of the non-pathological subject, on one hand, and for the treatment of the pathological subject (phobias, distress, depression, pain, tinnitus, etc.), on the other. The cognitive revolution is associated with a certain acceptance of responsibility by the subject: agreeing to suitable training and respecting the protocol thus learned offer the means to flourish and/or resolve a given psychological problem. It may be noted that cognitive psychology accompanied the major groundswell that transformed the disciplined society prior to the 1960s into a society based on the individual, capability, and responsibility. The question is no longer “What can I do to get better but am I capable of it ?” [15].

Behaviourism is clear regarding the origin of the unpleasant nature of tinnitus (i.e., Pavlovian conditioning between tinnitus and a negative event in life). On the other hand, the explanation is less clear for the cognitive model: if tinnitus is disagreeable because of erroneous beliefs, one might question the origins of these beliefs, which may prove disruptive to the normal course of life. One might even question whether the existence of negative beliefs is necessary for tinnitus to be attributed to a disagreeable character. Pain is by definition a disagreeable experience without being the result of a thought or an erroneous belief. Tinnitus can, like pain, be an experience that is necessarily negative, without recourse to negative automatic thoughts. Regarding hearing, it would appear that certain sounds are unpleasant independently of any cognitive construction (artificial sounds–pure sounds, narrow band noise, and high-frequency noise, for example) [10,16]. It is possible that the unpleasant aspect of tinnitus precedes thought and that thoughts are formulated a posteriori by the subject but they are not causal in making tinnitus a disagreeable experience. If the discomfort associated with tinnitus is only a matter of negative automatic thoughts and beliefs, then counselling should resolve the issue in all cases; however, this is far from the case. Nevertheless, the evidence showing that thoughts and beliefs can modulate the intensity of pain is abundant in the literature. The placebo effect may be seen as a phenomenon of belief that can modulate pain [17]. Erroneous beliefs certainly play a role in the lived experience of tinnitus, but the whole question is to what extent they intervene in the process that makes tinnitus a disagreeable experience and in what proportion.

The cognitive models are centred on the subject, their history, and their uniqueness, but paradoxically, they also objectivise and generalise to the extent that tinnitus as lived experience is ignored or minimally taken into account. In other words, cognitive behavioural therapies focus on the way to manage tinnitus, that is, on the reclassification of its emotive impact and on the detached attitude to adopt vis-à-vis an internal background noise that is, when all is said and done, harmless and common. But these therapies do nothing more than apply a standardised reference framework and take little account of the uniqueness of the symptom and of its phenomenology. They do not seek to elucidate the meaning of tinnitus for the subject, that is, what makes it “painful” and intrusive. And that is exactly what is lacking, in my view, in the psychology of tinnitus: a more thorough understanding of what is at stake with the emergence and the presence of tinnitus. In what way and how does tinnitus destabilise the subject’s peace of mind? What does the subject’s peace of mind consist of, and what can disturb it? To be more precise, one might wonder what is at play with tinnitus, and what major psychic mechanisms in humans can it reveal. The literature is relatively abundant, notably, phenomenological, on pain and suffering (or distress), but tinnitus has never been tackled with a similar approach. The aim of the present article is to address this lack. This reflection is undertaken in the belief that understanding these aspects may assist in the treatment of the patients.

## 3. What Is Phenomenology, ‘the Science of Phenomena’?

Edmond Husserl is considered the ‘father’ of phenomenology. Husserl wished to develop a rigorous method for apprehending and understanding the possibility of knowing or, according to Bordeleau [18], “(…) *giving actual human experiences, apart from their succession in time, an intrinsic truth or sense of origin*)”. Husserl’s phenomenological project was “*to go to the things themselves*”, that is to say, to go beyond what is revealed to the consciousness to apprehend how that is manifested. The question raised by Husserl may thus be formulated as ‘*How can I objectively know the world without denying my subjectivity (against empiricism), or hypostasis (against idealism)?*’. Phenomenology is often assimilated to a kind of ‘science of phenomena’, which aims to go beyond the dichotomy between empiricism and idealism. What are we to make of this? Husserl distinguishes the double meaning of the term ‘phenomenon’: ‘what appears’ (the form and content of appearance) and ‘the appearing’ (how it appears, which is the basis of the possibility of knowing). What is of the order of ‘what appears’ to an observer is of the order of empirical experience and prejudice, that is, “*the states of actual things, contingent on and part of an objective space and time*” [18]. Phenomenology, as a way of apprehending the world, is not limited to the appearance of phenomena, that is, to the simple subjective experience we have of them, according to a ‘natural’ mode of operation (the fact that believing that everyday empirical experience can afford access to the real world) biased by prior assumptions and beliefs. This aspect is rather the subject of study in positive and empirical science, which includes experimental psychology. The phenomenological approach, on the contrary, embraces the totality of the phenomenon down to its ontological core, its essence, since everything that appears is the appearance of something of the most essential and irreducible nature, or its ‘appearing’, in other words, what causes the fact that prior to what is perceived there is perceiving. Phenomenology is “*not a science of the particular content of what occurs, arrives or appears, but what makes the content of the fact, of the event and of the phenomenon, that is their pure phenomenality or their appearing (…). Phenomenology (…) analyses the layers of meaning, distinguishes them to take into account the level of human experience, relates them to each other, elucidates the how of the manifestation of what occurs, arrives or appears empirically, in order to reveal their originality*.” [18].

Husserl proposes a method to reach the essence of phenomena. Nonetheless, it is necessary to exercise caution when we talk of the phenomenological ‘method’, in that phenomenology seems closer to an attitude than a strict method with a systematic procedure [19]. This attitude, similar to the *épochè* (interruption) of the Sceptics and referred to as phenomenological reduction, is contrasted with a ‘natural’ attitude, or that of the everyday life of humans who live with their prejudices. It consists of suspending judgement, setting aside the natural world, and abolishing any prior assumption and familiarity, any model or abstraction, in order to accept phenomena, the lived experience, and the content of consciousness as they appear (‘the actual things’ or ‘the things themselves’) [20,21,22]. Husserl suggests that “*any state of consciousness in general is, in itself, consciousness of something*” [23]. This famous expression of Husserl signifies that consciousness always aims at a content, an intentional object (real or imagined) that it is always in tension with the world. This characteristic of consciousness defines what Husserl calls its ‘*intentionality*’, that is, the fact that it is the bearer of an intention, that it is open and directed towards something other than itself.

Phenomenology is used in the humanities, in general, and in the study of human consciousness, in particular. In this context, we might ask what are the differences between psychology and phenomenology, and what is added by phenomenology to what is already taken into account and described by psychology? Psychology (experimental), as a discipline examining the ‘what appears’ of psychic phenomena seen as objects, is limited to accumulating empirical facts enclosed within themselves. Phenomenology, on the other hand, takes no interest in the appearance of psychic phenomena or, at least, it only considers them to reveal “the totality of what is Man” [22], that is, the fact that s/he becomes aware of thinking something (s/he finds himself thinking of something). Like an archaeologist, the phenomenologist ‘digs’ beneath the habits and ‘natural evidence’ (empirical everyday experience) to find the deep structural determinants of the lived experience as it occurs. The return, ‘the things themselves’, is the attitude consisting of moving from the phenomenon to the subject (and not the reverse in the case of the realist’s position), which will permit the unveiling of acts of consciousness (thought, perception, imagination, will, affectivity, etc.), its original core, and the intention that drives it. Phenomenology is a descriptive analysis that reveals “*the necessary and universal structures of all lived experience*” [18]. Phenomenology should not be confused with simple introspection, as the latter only encounters the natural fact instantly provided by the consciousness (‘what appears’). The phenomenological process, with the method of ‘reduction’, that is, placing “the world in parentheses”, radically distinguishes it from introspection [22].

## 4. Sketching Out a Phenomenology of Tinnitus

The phenomenon that interests us here is tinnitus and the way it is experienced by the subject. According to the phenomenological approach, the way the subject experiences tinnitus is by its very essence meaningful; therefore, the whole purpose of the approach is to seek the meaning of this lived experience. The aim of the present article is to shed light on its ‘appearing’ (how it is what it is) beyond its appearance (‘what it is’ in terms of appearance). What the subjects report regarding their tinnitus is well known: “My tinnitus prevents me from concentrating”, “My tinnitus stops me sleeping”, “My tinnitus is driving me crazy”, “My tinnitus stops me living”, etc. [24,25]. What we see here is a description of the empirical lived experience, superficial and ‘natural’, in the sense of ‘what appears’ directly to the subject, and what is directly accessible to them. Yet, although what the subjects report documents a certain aspect of the lived experience and presents an interest, notably, in clinical practice, for the recognition of their complaint and for the choice of treatment, the symptoms listed tell us very little about the deep meaning of tinnitus, in particular, with regard to its impact on existence. My purpose is to use the phenomenological approach to reveal what is behind the apparent symptoms of the subjects and to find the ‘essence’ or the phenomenon of tinnitus. Using this research on the ‘appearing’ of tinnitus, my aim is, as was indicated by Sartre (1938), “*to go beyond the psychic, beyond the situation of Man in the world, to the source of Man, the world and the psychic*”, that is, as far as the original and intrinsic characteristics of humankind. At this level of description and comprehension, it will therefore be possible to establish a parallel between tinnitus and other symptoms such as chronic pain. A comparison between tinnitus and pain will enable us to further clarify this approach. Whereas tinnitus and pain are distinguished by ‘what appears’ (the sensation concerns different senses), the two phenomena could share a common ‘appearing’, and thus reveal some basic principles of human consciousness.

The first aspect of this phenomenological reflection is that tinnitus shows a resemblance to chronic pain in that it is continuous, formless, unpleasant, and indicates nothing about the world. Like pain, tinnitus is indeed a matter of ‘feeling’ rather than ‘perceiving’. Although certain authors have assimilated tinnitus to pain, this comparison is nonetheless limited to their physiopathology [26,27]. It is surprising that their phenomenology with regard to what they might have in common was not discussed at an earlier stage. I wonder, therefore, if tinnitus might be described as a painful phenomenon. To answer this question, I shall tackle the phenomenology of perception and I shall define in what way pain and tinnitus differ radically from perception. We shall also see how an unpleasant phenomenon that lasts may deplete the resistance of the subject, profoundly impact their being, and cause suffering. The second point developed concerns the central role played by the auditory system as a system of vigilance. An auditory phenomenon does not have the same significance as a visual phenomenon, and that is certainly not neutral when it comes to understanding the subjective experience of tinnitus and of phosphenes, the latter not being as serious a clinical problem (in terms of severity) as the former. The third important point is that tinnitus is experienced as an event, that is, it triggers a crisis in the subject. For that reason, tinnitus defines a temporality, a before and an after the appearance of tinnitus. The rapidity of its occurrence adds a dimension of brutality to its significance, rather like acute pain, which seizes the subject. As an event, tinnitus has a certain impact on the subject, and it will follow a certain trajectory [19].

## 5. What Is Perceiving?

Maurice Merleau-Ponty gave us a profound and inspiring reflection on perception in one of his major works, *La Phénoménologie de la Perception* [21]. The key concept developed by Merleau-Ponty is that of *corps propre* or *corps phénoménal* (phenomenological body). The *corps propre* is, among other things, a response to Cartesian dualism, i.e., a separation between the soul and the body, even if Descartes himself recognises that the two can be tightly coupled. Above all, it is a response to the view of a separation between a subject and the world. The reflection of Merleau-Ponty, like any phenomenologist, aims to go beyond the opposition between realism and idealism, by going back to the subjectivity of the subject to try to understand the ‘appearing’ of phenomena. We do not face the world, we are in the world, and we are embedded within the world. We are in relation with the world, and we form with it a system. We are in the world because there is a world that came before us. And there is a world because the body is capable of accepting it, of feeling it. The *corps propre* is that by which the subject is in the world, and they are in communion with it. The *corps propre* is our point of view on the world; it is the incarnation of living, which possesses “*its own life, its own rhythms of deployment*” [28]. Merleau-Ponty gives the following definition of the *corps propre*: *“The corps propre is in the world like the heart in the organism: it continuously keeps the visible spectacle alive, it animates it and nourishes it internally, with it, it forms a system*”.

The *corps propre* is not a closed interior where the subject observes the world. Perception is born of this openness to the world and the predisposition of the body to the world: “*Perception is not a science of the world, it is not even an act, a deliberate position, it is the basis upon which all acts derive and it is presupposed by them. The world is not an object of which I possess before me the law of constitution, it is the natural setting and field of all my thoughts and all my explicit perceptions. (…). When I return to myself from the dogmatism of common sense or the* dogmatism *of science, I find not a core of intrinsic truth, but a subject dedicated to the world”*.

The subject of the sensation, the being in the world, may be seen as “*a power that knows a certain milieu of existence or synchronises with it*”. What is meant by that is that perception is the result of the deployment of the body, of a certain strategy for exploration of the world, of a “coupling” of the body with the world [29]. Seeing the world is looking at it according to a certain dynamic and scanning it in a certain way. It is the same for the tactile senses, the hands ‘look at’ objects, they touch, manipulate, press, and slide over the object. Not only is the *corps propre* dedicated to the world, but it also ‘sympathises’ or ‘communes with’ it: “ (…) *what can be felt has not only a driving and vital meaning but it is nothing other than a certain manner of being in the world which is offered to us from a point in space, that our body takes up and adopts if it is capable of it, and sensation is literally a communion.* “ [21].

Using exploration, the perceptive process is deployed in time, it secretes a time: “My body takes possession of time, it brings into existence a past and a future for a present, it is not a thing, it makes time instead of enduring it” [21]. All perception is never complete or incomplete inasmuch as the object cannot be perceived at the same time in all its facets. The corps propre then achieves a passive perceptive synthesis that extends the perceived given to form a real and continuous object. Sitting at my desk, my gaze cannot see around the cup in front of me, but I extrapolate its shape, I postulate a continuity of the cup. It is in this sense that Merleau-Ponty wrote “Perception affirms more things than it can grasp” [30]. It is this going beyond perception, the filling in of the information lacking, that makes the object appear to us as outside and at a distance from the body.

Perception results from a specific exploration of the world, but it is not ‘given’ (like feeling). It is not an act of conscious and entirely individual perception that performs the synthesis of the object. In reality, we profit from a synthesis that has already been performed by means of “a certain overall assembly”, a “nexus of relations”, whereby the body adapts to the world. This is what Merleau-Ponty means when he says “*our body knows more than we do about the world*.” In other words, the thing “*is adopted within by us, reconstituted and experiences by us inasmuch as it is linked to a world of which we carry with us the fundamental structures and of which it is but one of the possible concretions*” [21]. *In fine*, “*the whole of nature is the staging of our own life or our interlocutor in a sort of dialogue.”* “*Nature is within”, said Cezanne. “Quality, light, colour, depth, which are over there in front of us, are only there because they awaken an echo in our own body, because it welcomes them*” [21]. The drawing and the painting are not a copy of the world, they are “*the inside of the outside and the outside of the inside*” [29]. In *Le Petit Prince* [31], the pilot of the plane that broke down in the desert spoke to the desert landscape: “*I’ve always loved the desert. One sits on a sand dune. One sees nothing. One hears nothing. And yet something radiates in silence (...)”.* This passage illustrates what Merleau-Ponty describes: what we perceive is the outside of our inside: the objects of the world seem to vibrate or radiate energy, and they are, in reality, animated by our tension in the world, the intentionality of consciousness.

The world is not an accumulation of independent sensations enclosed within themselves, the face of my interlocutor, his voice when he speaks to me, and the slight movements of his head, are linked to each other, they can be transferred from one to the other. The *corps propre* makes the unity of the object by living it: it is first thought in it where it resides and where it can be found, and the synthesis of this phenomenon enables us to experience it. It is well known, for example, that visual and auditory stimuli tend to merge, the sound conferring a vital presence on the images that they would not have otherwise.

## 6. Body, Affectivity, and Emotion

The body is so dedicated to the world that most of the time it is silent, forgotten, and transparent. Feelings tend to be experienced in a body without contours, which seems to vanish in the act of perceiving. The subject perceiving then has the illusion that his or her soul is floating or ‘ethereal’ in the body The default mode of the body is transparency. The effects, however, are experienced in a body in excess, a deformed body, a body which thickens. Our body indeed has this astonishing characteristic that it oscillates between two poles: on one hand, it vanishes in the act of perceiving, and, on the other hand, it becomes invasive in the affective processes [28]. In the affective experience of pleasure, the body appears to be filled up, and the unity between the body and the being is maintained or even enhanced. In displeasure, by contrast, the body is reduced to the region of the unpleasant effect while the unity between the body and the being is broken.

Affectivity is the experienced correspondence of the effects. If feelings allow the revelation of beings and things by being in communion with them, to the point of being confused with them, then affectivity gives us access to the world, in which manner there are beings and things. Affectivity transfigures beings and things into a whole coloured by a certain affective tonality, which might be described by qualities such as ‘luminous’, ‘transparent’, ‘clear’, ‘opaque’, ‘obscure’, ‘heavy’, etc. [28]. Affectivity is the repertoire of the artist, the writer, and the poet. The mood of the morning might be light, the atmosphere of a family reunion may be heavy, etc.

Emotion may be defined as a radical way of transforming the world from within by means of magic when the situation is desperate [22]: “*Thus the origin of emotion is a spontaneous and experienced degradation of the consciousness in the face of the world. What it cannot tolerate in a certain way, it attempts to seize in another way, by approaching the consciousness of sleep, dreams and hysteria. And the disturbance of the body is nothing other than the experienced belief of the consciousness inasmuch as it is seen from the outside*”. Emotion is associated with physiological reactions (acceleration of the heart rate, shivering, sweating, etc.), which not only prepare the body for action but are also perceived and amplify even more the serious character of the experience undergone. Emotion is a degraded state of consciousness, it is a bewitchment, a seizing of power by “an inner other”. Emotion manifests itself when the consciousness is without response to a situation (we just missed a train a few seconds before), and it then seizes the reins of the behaviour. Does not anger arise from a feeling of impotence and despair? Is not the goal of emotion, as Nietzsche suggested, to stifle one feeling of frustration by another: “(…) *one wishes to deaden a secret burning pain, that has become intolerable, by means of an emotion that is more violent whatever it is, and to drive out, at least momentarily, this pain from the consciousness* (…)” [32]. Emotion is paradoxical since it is what makes the subject behave with conviction and ardour while consciousness is diminished. When urgency requires it, when we are confronted by the imminence of a catastrophe, emotion hypnotises the consciousness and alters the trajectory of behaviour (in the register of flight or fight). Emotion is the feeling of a situation of extreme crisis, a disruption of our normal relationship with the world, which is then felt with excessive intensity. We shall see below the correspondence between emotion and pain. Emotion can also be deployed in certain contexts intended for an audience or for oneself. Emotion adds conviction to a given attitude, whether it be joy, sorrow, submission, domination, anger, etc. Do we not say, “if I had been any angrier, I would have acted it out”? Emotion, real or feigned, also has the function of finding a way out of a desperate situation. Bursting into tears in front of a jury, or openly capitulating in the face of an ordeal, is a solution for changing the attitude of the jury and reversing responsibilities. The jury is induced to behave as a consoler rather than an inquisitor. It would also appear that we would attempt to get ourselves into an emotional state to make use of emotion as an aid to decision-making. We bewitch ourselves to amplify and perhaps trigger a decision in a chosen direction: “I wouldn’t have chosen to move abroad if I had not felt excited at the idea of discovering a different world”. It is worth noting that emotion always presents certain inertia once it is triggered beyond a certain threshold. It takes a certain amount of time to get back to a state without emotion. This inertia is perhaps the condition for achieving a survival behaviour (escape, for instance).

## 7. What Is Pain?

We have seen above that perception is an active process resulting from an attitude of openness and communion with the world [21]. Pain, in contrast to perception, does not derive from an active process of exploration of the world. There is not a ‘pain sense’ as there is a visual or tactile sense, with an organ and a strategy pre-adapted to the world: “*Pain does not objectively elucidate anything and its role is just to make us aware of the ill by an immediate feeling*”. (Pradines, cited in Buyttendijk, 1951 [33]). The only thing that may be indicated by pain is a certain localisation within the body (and still, the pain location does not always correspond to the location of injury). Pain abruptly closes the relationship between the openness of the body and the world; it involves a breach between the body and the world. It places the horizon of the *corps propre* at the level of the physical body: the subject no longer has the capacity to deploy towards the world. In summary, whereas perception is openness, expansion, and fusion with the world, pain is, on the contrary, a rift, a closing down, and a withdrawal.

The sensation of pain is never really over. It always conserves a mysterious character and is worrying in that it always remains without shape. The cry (the non-verbal sound that signals a stunned state) is to hearing what pain is to the body, “*a force that plunges in a whirl into a bottomless gulf*” (Plessner, cited in Buyttendijk, 1951 [33]). Whereas perception is shape, proportion, measure, and even, in certain cases, beauty, pain is shapeless, excess, and monstrosity [34]. Pain is the opposite of order; it is the sensation of the destruction of the ordered relationship between the body and the world.

Pain has no object (in contrast to perception), it only has a subject who experiences it. There is thus no distance between the subject and the sensation of pain that fills consciousness. Pain deforms the representation of the body, which then ceases to be transparent: it becomes darker and thicker. The phenomenon of pain is accompanied by a swelling of the body at the pain location. The world is limited to the place on the body that the wasp just stung: it becomes swollen with its pain, which seems to erupt excessively.

## 8. The Affective Part of Pain

The difference between pain and perception is not a simple difference in intensity but a difference in nature [35]. Pain is an eruption, an excess: “*Pain is the sensation of a situation of tension and crisis, the sensation of the abolition of our normal relation with our body, which is attacked in one of its parts, but with maximum intensity, like our own reality injured, while we are incapable of any adequate reaction*” [33]. There is excess in the sensation of pain, which accompanies the piercing of a needle, the heat of a fire, and the crushing of flesh and bones. This something in excess appears to be an affect (pain affect), a kind of conviction (of the order of conscious intuition), which is added to the physical sensation (pain sensation). We note that the reactions associated with pain resemble those of anguish and anger. The swearwords that we emit during pain are from the repertoire of the affects and emotions. We note that the swearwords in return act on the perceived pain, and it has been shown that they may be able to reduce it [36,37]! We swear when the hammer injures our finger. For what reasons? Here, again, the verbalisations issued from the register of anger and insults are an attempt to give oneself the illusion of taking back control of the situation. Pain is interpreted as a violence (in French, we speak of ‘*rage de dents*’ for toothache), and we respond to the feeling experienced with another feeling. The affective part of pain may thus be seen as a phenomenon of belief or faith, comparable to emotions such as fear or anger [22]. It adds affect (gravity and seriousness, as Sartre suggested) to the sensation of pain, that is, that in which (how, in what way) the sensation is experienced. Pain is the sensation of an emergency, an imminent disaster, or a state of panic affecting the body. The pain affect gives pain its character of tension and crisis, thus affecting the subject in a situation of survival. As a phenomenon of ‘unconscious conviction’, the pain affect possesses a link with beliefs and, for this reason, it can be manipulated by them. There are numerous examples showing that pain may be reduced and sometimes even suppressed according to the context or, in other words, according to the meaning that is attributed to it [17]. What we call the placebo effect would thus be a (calming) belief regarding the pain feeling. There is abundant literature reporting the effects of context on the lived experience of pain [38,39,40,41,42]. Uncertainty regarding the cause of pain tends to increase its severity, whereas removing uncertainty calms pain. Along the same lines, the effectiveness of shamanistic healing might be explained by this type of phenomenon of belief [43]: an ailment of the body whose manifestation (symptom) seems abnormal is cured when this manifestation is related to the ‘cosmic’ order seen as normal and intelligible by the subject.

## 9. From Pain to Distress

Pain is usually experienced as a transitory phenomenon, that is to say, it disappears when the source of the ailment has disappeared. It is on the absolute condition that this promise is kept that we consent to tolerate it. There are, however, situations where pain lasts, where the injury or irritation to the tissue persists or even where there is no lesion that is visible or detectable using the means available to modern medicine (neuropathic pain). In this case, pain is not accepted as a transitory incident that will soon be forgotten but as a tragedy, if the intensity of the pain is sufficient, which might well take over one’s whole existence. Pain that lasts creates a barrier to life. This type of pain is experienced as a wave that sweeps over subjects, it submerges their consciousness, at once drowns and imprisons them, leaving them powerless. Seized by pain, the subject becomes incapable of projecting into the world, of making plans, and of living in it. The subject may put up resistance to his situation with an issue that is inevitably ill-fated as long as the pain persists. The subject engages in a struggle where the outcome is a certain failure at the cost of inexorably depleting his resources. Offering no respite, chronic pain tends to overwhelm the subject.

During this period when the subject resists and battles, emotion plays an important role. It is a spontaneous response of consciousness to put some heart into the struggle against the ailment; emotion motivates subjects, supports them, encourages them, and lends them conviction. But the recourse to the register of the emotions (and of the activation of the sympathetic system) can only be a temporary response because of its intensity. Having recourse to the emotions is the choice of ‘going for broke’. Like pain, emotions cannot last because they are a radical response to a desperate situation, they drain the subject’s resources, both physiological and psychic. Like pain, emotions change their nature when they last, they wear down subjects, consume their resources, and end up exhausting him.

Distress or suffering arises when subjects lose hope for a positive outcome to their battle. Suffering is the feeling experienced by subjects when they can no longer cope with the situation and when they become aware of their vulnerability. Suffering is an ailment that digs much more deeply into the being than the body, reaching the core of existence and identity. Pain and suffering do not impact the same psychic levels in the subject. As proposed by Paul Ricoeur, “*The term pain (will be reserved) for affects felt as localised in particular organs of the body or in the whole body, and the term suffering for affects relating to reflexivity, language, the relationship with the self, the relationship with others, the relationship with meaning, with questioning*” [44]. Pain that lasts is absurd; without response, without any issue, it disorganises the subject in their whole psychological structure. The assault is this time situated not only at the level of the relationship between the being and the body but in the depths of the very being, of existence in its temporal deployment (man is “the being of the eternal future”, as Nietzsche said). For the suffering subject, existence itself is impossible, life can no longer be lived, there is no perspective, and everything comes down to pain. Like pain, which is an incomplete sensation, suffering is always an incomplete thought. What the subject took as normal, that is, a life going on in a silent body without existential questioning, falls apart: by losing illusions, the subject loses the natural evidence of living. Suffering is the moment when man encounters the meaningless and the void, in other words, when one questions the meaning of existence with the greatest acuity. The loss of natural evidence signals the end of a certain carefree progression of existence. Whatever happens, life will have to continue in a different way, and the subject will have to change: “*Only great pain can ultimately overcome the mind by teaching it the great suspicion. This long, slow pain which consumes us little by little, which takes its time, and forces us to delve into our deepest recesses, and strips us of all trust, all goodwill, all blindness, all indulgence and the mediocrity where we had previously placed our humanity. I doubt whether such pain makes us better, but I know that it makes us more profound”* [45].

Suffering upsets the relationship with time [34]. Perception is indeed deployed within a time defined by the body itself with its exploration of the world. Time is “secreted” by the body in its relationship with the world, “*it makes time rather than being subject to it*” [21]. Existence is not deployed ‘within’ time but deploys (makes exist, “secretes”) time [46]. Conversely, we described pain not as a perception but as a sensation that is characterised by a body withdrawn into itself and closed to the world. Suffering imprisons a being in the eternal presence of their pain, it freezes any attempt at making plans, and it thus profoundly disorganises the deployment of the subject in life. The present is a prison where the subject remains captive as long as there is no future or, rather, the future is only a repeat of the present. That is the whole problem of chronic pain, pain in the present is repeated by the pain to come that we anticipate since nothing can stop it. We suffer as much if not more from the pain to come, from the certainty of the continuity of the pain, the lack of respite, as we suffer from the pain experienced in the present. Thus, chronic pain is not only associated with the abolition of the distance between the subject and the sensation but also with a suppression of the ‘distance’ between the present and the future, inasmuch as the future is the unflagging repetition of the present.

In suffering, the consciousness fails as it cannot find a satisfactory response to the impasse the crisis represents. Without the capacity to make plans, Man cannot exist. What do we mean by exist? The reality of humankind is that we are beings incapable, according to Pascal’s expression, “of remaining in repose in a room” (cited par Porée, 2020). In other words, existence is an “ecstasy” that is a tension or a movement to open up to the world and leave the self: “*We exist first and foremost and most often out of ourselves*” [46]. In short, there is no suffering without a dogged consciousness that ‘repeats’ the present pain by obstinately seeking a solution to resolve the crisis. The suffering of the subject who withdraws into themself is the result of an abnormal psychological tension when the consciousness has no way out and turns on itself.

Suffering is an absurdity, but if it “*is in a way without ‘purpose’, it is not without a ‘why?*’” Suffering is not only felt but is also questioned, interpreted, and judged. “Why me? Why my loved one? Was I chosen for suffering?” Suffering is interpreted as a manifestation of evil (*‘mal’*) [44]. It is worth noting that the French word *‘mal’* expresses at once pain (*avoir mal*) and an act contrary to morality (*faire le mal*), as if there were a relationship between suffering and wrongdoing. Suffering shares with wrongdoing the fact that they are present in the world but they should not be. “*Whence the question: why does what should not be, exist? The moral question here becomes a metaphysical question*” [44]. This question was addressed, notably, by Christian thinkers: if a god who is good and almighty exists, how can we explain the presence of evil in the world? The Book of Job in the Old Testament presents this question in exemplary fashion: Job is very pious, but he is put to the test (by Satan with God’s agreement) and loses his goods and his children. Job’s friends could not believe that Job did not commit sin since it was unthinkable that God would allow Job’s suffering if he were pious. Job committed no sin and did not deny God. Job was finally rewarded by God for his loyalty. It might be believed that the association of wrongdoing and the suffering was in reality just a Judeo-Christian invention, but this is not the case. In the myths of the origin of the world, the emergence of death and disease were very often the result of divine punishment in reaction to wrongdoing (transgression of a taboo, betrayal, etc.) [47]. In other words, humankind has raised the question of the reason for pain, death, and the meaninglessness of pain; “what is and should not be” is not accepted as such but is interpreted as a flaw in a mythical and imaginary world. People of an earlier age, like people today, never looked at the meaninglessness of existence “right in the eyes”. No doubt they never wished to avoid falling into an abyss of doubt. Or could they simply not conceive of meaningless? What the Greek myths, in particular, repeat incessantly are the causal relations: complex, indirect, and unexpected, which are beyond the grasp of powerless and credulous protagonists. The heroes of the myths are the puppets of destiny, and it is often thinking to escape it that they hurtle towards it. The myth is a way of making intelligible what is beyond people in their own lives, by presenting existence in the form of a metaphor. The myth enables humankind to represent the meaningless, to experience it, and to practise it to better tolerate it. Confronted with suffering, we return to an original and profound questioning of our own origins and the meaning of life.

## 10. From Pain to ‘Impressions of Pain’

Above, I discussed pain related to the physical body. One might wonder whether a ‘painful phenomenon’ or an ‘unpleasant sensory phenomenon’, which is not limited to the physical body (tactile sense) and that shares a certain number of the characteristics of pain, might exist. The characteristics of impressions of pain are the sudden eruption of the sensation, the character of urgency and intrusion (the sensation of being penetrated by the sensation and overwhelmed by it), and the increased vigilance and closing to the world. According to Buyttendijk, the senses that impressions of pain can transmit are senses that act at a distance, that is, sight and hearing: “*These are the organs of an ordered relation with the outside world and, in consequence, the only ones capable of enabling an aesthetic development. What is unpleasant is the disturbance of the aesthetic order (…) We are then impacted, injured, disorganised and–what is essential–delivered defenceless to meaningless incoherent destruction deprived of any coherence. Like pain, optical and acoustic sensations are imposed on us; they cause us meaningless discomfort and irritate us, like real protopathic impressions*” [33].

It would appear that hearing is more capable of producing an impression of pain than sight. Although noise pollution, as well as visual pollution (advertising hoardings, urbanisation, wind turbines, etc.), can be defined, the impact of sound nuisances is considerable and is far greater than that of visual nuisances. It is possible that this ‘superiority’ of the auditory over the visual may be explained by the physiological integration of hearing within the emotional system. Furthermore, hearing does not imply the presence of an object as vision does. A sound is a physical event (see below), but it is not necessarily identified or localised with precision, and the source may be unidentified. Sound reduces the distances and ‘touches’ the body, which makes hearing comparable to touch. These two senses have in common a strong sensitivity to intrusion, much greater than that related to the sense of sight. A sound may be the vector of an emergency or an alert with cries and screams. These vocalisations of a particular type [48] suddenly erupt like pain and agitate the subject with maximum intensity. Like pain, a cry opens a breach in the orderly arrangement of things; it is the incoherent element that destroys the harmony of the world.

Certain sounds are unanimously recognised as being unpleasant, regardless of the context of their occurrence. The sound produced by chalk on a slate blackboard, distorted harmonies, modulations of the amplitude in the rugosity range (between 20 Hz and 300 Hz), or pure sounds at high frequencies are perceived as unpleasant [10,48,49]. There are also clinical conditions in auditions that are specifically defined by a drop in the threshold of tolerance to sounds. Subjects presenting hyperacusis are particularly sensitive to the loudness (subjective intensity) of sounds [10,50]. Subjects presenting misophonia, on the other hand, are disturbed by specific sounds known as ‘triggers’, which cause an intense emotional reaction. These sounds are usually the sound of mastication produced by human subjects [51,52,53,54]. In misophonia, the subject cannot stem the emotional reaction caused by the sound that penetrates and overwhelms them. The trigger sounds seem to breach a certain private or personal sphere and may be experienced as an assault in the same way as pain. In short, the disturbance and the oppression linked to sound may be so profound that it may disrupt the normal course of life (i.e., the relation of the subject to the world). There is no visual equivalent of the cry.

If pain signals a severe and immediate assault on the body, then the impression of pain is a less intense attack on the normal relation between the subject and the world. This alteration may be moral or aesthetic, but it may also be of the order of a reflex. The impression of pain may, depending on the severity, go so far as to occupy the forefront of the consciousness. As described earlier, the negative experience of pain depends on the context and also the person’s attitude. The same is true for impressions of pain in the senses. On the other hand, the character of the pain, in other words, the sensation itself, the depth of the ‘injury’, the violence, and the upset, are nonetheless less intense and ineluctable than pain issuing from the physical body [33].

## 11. What Is Hearing?

### 11.1. Hearing Is the Channel of Vigilance

As we saw earlier, the body forms a system with the world, and thus its properties depend on the laws of physics. To describe it in the manner of Merleau-Ponty, it is as much the sound that makes the auditory system as the auditory system that makes the perception of sound. In other words, there would not be a sense of hearing if there were no sounds in the world (with their characteristics), and there would not be a world of sound without a sense of hearing to take them into account. In this context, it is useful to recall the fundamental notions of sound physics. Hearing is above all the sensory channel that enables the detection of events in the world, that is, the phenomena associated with physical movement. Sound is produced by the vibration of an object, which then emits a sound wave. This is transmitted from one element to the next by the movement of the particles in the environment throughout the space surrounding the vibrating object. A predator approaching prey cannot help leaving a ‘sound trace’ when it steps on a branch that breaks. This sound trace betrays the predator’s presence and is often enough to warn the prey of imminent danger. Thus, from the outset, sounds reflect the ‘sound trace’ of an event (the breaking branch) rather than the cause of this event (the predator). It is the vision, through looking in the direction of the event, that will objectivise the scene to conclude in a possible relationship of cause and effect (a branch has been broken by a predator). It is worth mentioning that precision in localising a sound source is very closely correlated with visual acuity [55]. This suggests that one of the roles of hearing, which has no vision, is the localisation of the source of the sound, and this is to indicate to sight where to look, which completes the objectivisation of the world.

A sound translates a causal sequence (the animal has broken a branch), a sort of living presence embedded in matter. In contrast, when a silent object is looked at, one may feel a certain unease: the object ‘radiates’ a certain tension and yet, paradoxically, it is inanimate, and it lacks the breath of life. Is that what Cézanne felt when he painted Sainte-Victoire Mountain, near Aix-en-Provence? It stands alone above the surrounding fields, its sides are steep slopes, its summit is more than 1000 metres high, and it gives off a cosmic energy because it is silent. The ancestor of music was perhaps giving life to objects by making them ‘speak’ one way or another, by blowing into a hollow bone, by making the string of a bow vibrate, or by tapping with a piece of wood or bone on a rock (some rocks can emit periodical sounds, presenting a certain pitch, and thus resembling voices) or on stretched skins. Hearing is a sensory channel which is subject to the world, much more than it results from a free exploratory process, like vision, for example. We might say that the act of hearing is performed by a movement that goes from the world to the subject, whereas the movement is the opposite in vision (see the section on the phenomenology of perception). Hearing has no viewpoint like vision, the sounds ‘affect’ us much more than we affect them.

In the course of evolution, hearing has acquired a major role, which is that of a system of vigilance wide open to space, and capable of detecting acoustic events that potentially herald danger, a bit like radar. Of course, hearing has acquired many other functions, including being the preferential channel of communication. In humans, in particular, the voice references a person and not an event. But I shall limit myself here to discussing its role as an alarm system. Hearing is the preferred channel as a system of vigilance because of the remarkable properties of sound. Sound travels fast (rapid detection), over long distances (the danger may be far off, but it is still possible to avoid it) and can pass through matter (which is useful for detecting sound events in dense milieu, for instance, in a forest). In addition, sounds may be detected whatever the position of the head in relation to the source of the sound, which is again an advantage compared to vision. Vision only sees what is looked at and upon which the attention is focused (magic exploits the incredible blindness we are capable of when the visual scene is complex and rapid). Finally, the auditory channel never closes given that there is no closure system (“it does not have eyelids” [56]). The physical properties of sound and the auditory system thus confer on hearing its role of radar, establishing the map of events in the world. Hearing often completes vision in the construction of a sensory event: at the moment of hurrying across a wide avenue, we hear a car coming and yet it is outside our field of vision since we are looking at the opposite side of the road. In most environments, and especially those that are ‘dense’ (urban jungle, dense forest, etc.), hearing enables us to establish a map of the events of the world well beyond the field of vision. The ‘golden ears’ of submarines play exactly the role of radar, without the assistance of vision, by detecting (and even identifying) the presence of other ships. We may note too that hearing enables us to estimate the distance of the object and sometimes its speed (Doppler effect).

Thus, hearing is a sense that is constantly open, which has no look, and which is ‘affected’ by a physical stimulus that travels at great speed and that nothing can stop. The auditory channel, more perhaps than any other sense, cannot offer protection against intrusions; it is at the mercy of all the events in the world, and one might say it is ‘exposed to any wind’. We can close our eyes or look away, avoid tactile stimulation, and hold the nose, but we cannot prevent a sound from penetrating the auditory system.

So far, we have seen that the auditory system is well placed to play the role of lookout, but it should in addition possess two supplementary characteristics: attracting attention and preparing for possible action with flight or confrontation if the situation calls for it. The auditory system does indeed present these two fundamental properties: sounds are very effective for attracting attention and triggering a physical reaction in the form of an activation of the sympathetic system and/or a contraction (reflex or not) of certain muscles [57]. In this sense, one might say that tension is permanent between the body (open to events, as described above) and the world of sound. The architecture of the body is so made that the auditory sense is intimately connected to the emotional system and the motor system. The auditory ‘radar’ is furthermore never completely at rest: the incoming sounds continue to be processed during sleep. The physiological reactions linked to the sounds (increase in the heart rate, for example) are produced during the period of wakefulness but also during the phase of sleep. These reactions diminish with repetition but do not completely disappear (partial habituation) [58].

### 11.2. Controlling the Auditory System

If the design of the auditory system is adapted to detect unexpected dangers and avoid them, then it may also be understood that the power of this system may constitute a risk for the normal course of life. It is rather like the immune system: it plays a role of warning and protection vis-à-vis agents outside the body, but it may be devastating if its response to an intrusion is too strong or if it turns against the body itself. Certain sound situations have the power to control consciousness and diminish the sound event. When these feelings are negative, the sounds may be experienced as real pain or torture. If such acoustic situations persist and/or are repeated, they may be experienced as turbulence and disrupt the tranquil course of life. In extreme cases, the subjects experiencing these situations may cut themselves off from the world, similar to the situation engendered by chronic pain. This is what happens in the case of noise nuisance, which is known to have a devastating effect on the quality of life [59]. The solution is often to move house if the acoustic environment becomes intolerable. It is not unusual that noise nuisance results in dramatic outcomes [60]. This ‘dark power’ of sound, by its intrusive and disagreeable character, may make auditory perception a painful sensation.

Hearing has the power, no doubt more than sight, to affectively colour the world. If sensations can reveal beings and things, the affect, in particular, that related to the acoustic environment, confers an affective colour on the world, that is to say, the way (or how) there is a world [28]. Music, for example, is a powerful means to confer a certain affective tonality on the world. In this sense, we give ourselves up to music as we give ourselves up to a belief. The music we listen to enables us to choose our mood and thus regulate it according to the desired tonality. Furthermore, whereas a visual landscape (peri-urban business zones, for example) may perhaps trigger intense aesthetic displeasure (the visual landscape is awful), a soundscape has the power to trigger intense, intrusive, and negative affects (the soundscape assaults the subject, brutalises them). While the first may, like it or not, be tolerated, the second may not be and may trigger a crisis. The sound reaches us and ‘stirs us up’ inside. A visual stimulus cannot compete with this power of sound. Let us illustrate what we just said about the link between hearing and affect by taking two different situations in which auditory sensations have an unpleasant character.

In the first example (borrowed from Daniel Tammam, 2007), let us imagine that we are driving at high speed along a very busy road in the morning to get to work. Our attention is distracted, we are mainly thinking about our day, while we leave the responsibility for driving to semi-automatic and unconscious processes. It is a morning like hundreds of others, our body is transparent, we have a feeling of confidence, and the world seems obvious to us. Suddenly, a strange noise emerges from our car, an abnormal noise that we are hearing for the first time. What is the cause of the noise? Has a tyre burst, or has the car hit an object that might have damaged some part of the car? Is it serious? Should we stop to check the state of the car, to make sure that there is no danger? Our anxiety is made even worse as stopping the car would be difficult if not impossible because the road is narrow and the rush-hour traffic very heavy. Suddenly, our world has tilted from the natural state of things to uncertainty: everything that was routine has become worrying, disturbing, and alarming [61]. Our consciousness is now entirely preoccupied with this noise; it is seized by the event in anticipation of a possible imminent disaster. There can be no question now of daydreaming as we examine the possible variations in this noise from every angle: “Is it louder, is it different?”, asking questions that are an attempt to understand if the possible problem is serious and if it is getting worse. This abnormal noise which appeared suddenly is comparable to pain that erupts: it saturates the consciousness, takes over the cognitive resources, it possesses a negative feeling (possibility of imminent disaster) and a dimension of urgency, it breaches the natural state of things, and it raises questions regarding the detailed functioning of the car. What is painful (or to a lesser degree, what is disagreeable) “*reminds us of what nature tries to make us forget, for our own good, for our peace of mind: the organic thing, infinitely heavy which is its fundament...*” [61].

Let us imagine another situation, that of an unpleasant noise devoid of the urgent character of the preceding analogy. We are on the same motorway, in the same car, with the same driving conditions. Suddenly, a noise emerges from the car, it is a high-pitched whistling. We know this noise well since we already heard it several times. It is the bonnet of the car that is not properly closed, which causes turbulence in the airflow. We try to ignore the noise, after all, it is not very loud, and above all, it does not indicate any imminent danger. The problem, however, is that it disturbs this moment when we would have liked to enjoy our reverie in peace. The noise grabs our attention, and it puts us in a state of tension and hyper-alertness despite ourselves. The noise is therefore unpleasant and annoying, but we put up with it and promise ourselves to take care to think of closing the bonnet when the car stops soon, as we have almost arrived. In contrast to the ‘painful’ previous situation, this situation is more like a ‘sound nuisance’, which generates an unpleasant but not urgent situation, and, above all, which is familiar and transitory (because the problem can be solved). The car noise gives rise to a certain disorder in terms of our relationship with the world and our expectations, but it is far from having the characteristic of imminent disaster in the first situation.

We used these two examples as metaphors for two different states relative to tinnitus. The first example (the possible breakdown, which was noisy and obscure) illustrates the emergence of tinnitus in a subject who is devastated by their tinnitus and by their lack of knowledge about tinnitus. The subject soon panics and seeks an explanation since s/he knows nothing about this new turbulence, this open breach in the harmonious order of things. By default, s/he envisages as many scenarios as possible and goes so far as to imagine the worst. The situation thus indicates an irregularity in the world that no longer allows consciousness to exercise its silent control ‘by default’. The subject then resides in the world of anguish, where a disaster may occur. The second example illustrates tinnitus that is better tolerated, which does not have the sinister and anguishing character of the previous one. Tinnitus is indeed an anomaly; it may be unpleasant, but it is part of the world, and it does not have the character of the imminent accident in the previous case. This tinnitus may be that of a subject who has finally learned enough about tinnitus to put it in the back of their mind, like the background noise, painful but common, of a mild case of sciatica.

## 12. What Is Tinnitus?

### 12.1. Tinnitus Is of the Order of ‘Feeling’

As I explained above, perceiving is an active process that implies an intention. An object is perceived from a body that opens to the world and communes with it. The intention and the means of perceiving pre-exist the subject; it is in this sense that Merleau-Ponty affirms that the body knows more than we do about the world [21]. In this context, tinnitus is not the result of an ordinary perceptive process, in that it is not the result of an exploration–objectivisation of the world: it does not obey the common rules of perceptive synthesis. To put it in phenomenological terms, there is, on one hand, the tinnitus characteristics (“that which appears”, at the surface of the phenomenon), which may be described in terms of pitch, timbre, and loudness, and, on the other hand, “the appearing”, which is not that of common perception. There is no distance between the tinnitus and the subject: it covers the distance that habitually separates the subject and the object, it is not the result of an examination of the world of tinnitus, and it is given in one block. Tinnitus does not have the appearance of the reality of an object in the world, it has neither outline nor shape, and it means nothing. It always conserves a certain incompleteness and mystery. To put it another way, there is no object in “the appearing” of tinnitus but only a subject who has an auditory experience. To summarise, tinnitus is thus of the order of feeling and not of perceiving, in the sense that it is given, without distance or elaboration, before having been the focus of an intention.

When tinnitus appears, it falls short of any cognitive elaboration or construction. It is for this reason, among others, that cognitive models are at best incomplete and at worst deceptive to account for the phenomenology of tinnitus. The discomfort of tinnitus is not only related to a negative thought that has become attached to the percept in a second phase of its emergence, a posteriori of a perceptive process that is, after all, quite ordinary. As indicated in this paragraph, “the appearing” of tinnitus infringes the usual rules of perception: it is immediately an aggression and an anomaly, like the cry or the pain that affects the subject and abolishes the distance between the self (the body) and the phenomenon. The important point here is that the “uncontrolled” appearance of tinnitus may upset the normal and organised relationship with the world and induce turbulence in the course of life. This original aspect of tinnitus is added to thoughts, beliefs, and counterproductive behaviours, making tinnitus both singular and distressing.

In terms of how it is felt, tinnitus veers towards ‘formlessness’ or ugliness: it sometimes resembles a pure sound, sometimes a sound that has a more or less broad spectrum, but never a ‘natural’ sound existing in the ‘physical’ world. Tinnitus is thus similar to a noise in the sense of a sound for which we do not know the source or an undesirable sound, which is one that is present but should not be or whose characteristics go beyond a certain norm of tolerance (too loud, too high-pitched, etc.). Yet, as Jacques Attali stressed, noise is interpreted as something negative: “*Noise has always been considered, in all cultures, as a source of destruction, of disorder; like a contamination, a pollution, an aggression. It is related to the idea of a weapon, a blasphemy, a scourge*” [62]. Just as a formless and unknown noise may suggest a disturbance and the imminence of disaster, tinnitus takes on a character of imperious urgency. Tinnitus then causes concern or anguish (see the example above for the mechanical car noise). More generally, whatever the general impact of tinnitus on the person, and even if tinnitus has only a minor effect on the quality of life, subjects always report that they prefer to live without their tinnitus. This is what was reported by Richard Tyler [6]: “*I have seen many patients in the clinic over the years who report that they are “not really bothered” by their tinnitus but paradoxically are anxious to know if there is a pill or something that will make it go away.*” The unpleasant nature of tinnitus is therefore not necessarily linked to a cognitive construction.

### 12.2. The Implications of Chronic Tinnitus

If tinnitus is a recognised public health issue, then it is because it impacts the well-being of subjects in a significant way. Yet, the diminution in well-being associated with tinnitus is very closely linked to the fact that tinnitus is a chronic sensation. Our sensory existence, composed of somatic, olfactory, visual, and/or auditory sensations, is permanently traversed by a ‘sensory background noise’ that may be more or less long-lasting. Who has not experienced itching, pain (headaches, sciatica, muscular pain after sport, blisters, grazes, etc.), phosphenes, olfactory illusions, or transitory auditory sensations in the form of high-pitched whistling? Our existence is, in reality, full of minor ailments of all kinds. We generally put up with them on the strict condition that they are transitory (of a duration not exceeding a few seconds to a few hours) and of tolerable intensity. In a certain way, this ‘sensory background noise’ is assimilated with normality as long as it does not exceed a certain threshold in terms of duration intensity. Here, we may refer to the distinction made by Canguilhem between anomaly (physiological variation in relation to a statistical norm) and abnormal (deviation vis-à-vis a norm of life of a certain value) [63].

Tinnitus, on the other hand, is an anomaly that lasts. It is an irregularity that becomes the rule and disturbs life. From a phenomenon that is disagreeable but inconsequential with regard to the quality of life when it is very brief (e.g., less than a minute), even if it is relatively frequent (e.g., fewer than five times a day), tinnitus may trigger a crisis if it is constant or quasi-constant. Constant tinnitus possesses an ‘authority’ that transitory tinnitus does not have. It is one or the other, either it signifies an organic problem that needs to be dealt with or it reflects a certain arbitrary defect of the body that has no significance; otherwise, the body can permanently generate an abnormal sensation. In both cases, tinnitus disrupts the habitual silence of the body, which is its default mode of functioning.

When the subject becomes aware that their tinnitus is constant, what was until then subnormal and unpleasant but common and without impact becomes abnormal and potentially impacts the quality of life. The subject is then challenged, possessed, and plunged into a state of helplessness and heightened vigilance. Tinnitus can maintain the subject under tension and give rise to a permanent state of unease. The subject does not have possession of their tinnitus, the way one habitually possesses an everyday object that one attempts to objectivise. Tinnitus erupts into the subjects’ lives inexorably and uncontrollably, and it remains present without any lapse. This aspect relates tinnitus to a “hallucinatory event” [64], and that is probably important when the subject (not delirious) assesses their tinnitus and gives it a meaning: the raw character, implacable and imposed, of tinnitus makes it an inquisitor.

The time frame of tinnitus changes its status, deprives the subject of a moment of respite (there are no eclipses in tinnitus), and alters the relation to time and to plans. The subject not only complains of the tinnitus experience in the present but also of the tinnitus experience that they will inevitably have the next day, the day after, etc. The problem is not so much tinnitus, which would be quite tolerable if it were transitory (the loudness remains low to moderate, at least in the majority of cases [65]), as it is the absence of respite and control. There is a kind of repeat in “the appearing” of tinnitus that plays a role in the subject’s suffering: it is experienced in the present and at, the same time, anticipated in the future. There is a two-fold movement in the sense of amplification of the worst, from unpleasantness in the present to despair in the future, since tinnitus offers no escape route. The battle against tinnitus is in vain: the subject is locked in and condemned to endure unremittingly the repetition of the present. Tinnitus thus resembles torture. In that sense, the suffering of an animal is not the same as that of a human. An animal always suffers in the present, and pain does not cause an existential crisis for it.

### 12.3. Tinnitus and Event

An event combines both the notion of surprise and that of ‘crisis’, in that it is a disruption in the course of life. An event may be a happy one (positive) or an unhappy one (negative), but it is by definition not neutral (otherwise, it would not be an event). The sudden death of a dear one, for example, is an event that is unexpected and triggers an existential crisis (‘How to cope with this loss ?’). A wedding is also an event, even if the preparations have been long and painstaking since such a moment can only be experienced in the moment and can always generate its share of surprises. In general, human beings seek surprise, uncertainty, and events that are significant and memorable. If by default a person’s life is contained within relatively tight emotional constraints, then they pursue in parallel a quest for intense feelings or passions: this is the role of shows of all kinds (we are spectators: theatre, cinema, sport, etc.) and certain activities (we are in action: profession, art, sport, social life with family, friends, clubs, etc.). The event is an episode of life to excess, a moment that is a landmark in the course of a life. An event, by definition, cannot be planned or anticipated: it delivers us, in a way, from our tension with the future. The event, which is associated with a heightened sense of existence, is lived in the present. The event frees us from speculation, and from the weight and the anguish of the future.

An event is first and foremost in the register of feeling before being in that of perceiving since it is delivered as is without intention. There is not necessarily any causal relationship between an event and the experience of it. The latter point is illustrated by Maldiney [66] in an example borrowed by Jérome Porée [19]. The event in question consists of a pedestrian who is run over and killed by a car. Among the people who go towards the victim, there is an experienced doctor and a young man who is confronted for the first time with a violent death. While the doctor calmly and methodically does what the situation requires of him, the young man is tetanised and shocked by the violence of the event. We might add that as concerns the doctor, the event concludes with the sentiment of duty accomplished, of having been capable of handling the crisis caused by the accident. For the young man, on the other hand, the accident triggered a crisis that is not concluded by the arrival of the ambulance or by the removal of the body, nor by distance from the event. The accident has in that instant devastated him and frozen him in a state of shock, being incapable of action. Thus, the same event (a violent accidental death) was the cause of two different and two radically divergent trajectories. The doctor was strengthened by the event: he showed his sang-froid in a situation that was urgent and difficult. The young man, in contrast, emerged weakened by this episode of crisis. His ‘impressionable’ reaction at the moment of the accident gradually developed into a heightened sensitivity. Certain smells, certain images, and certain noises acquired a persistent negative character, difficult to forget. He might also have a feeling of guilt for not having reacted as the situation required for the well-being of the victim, added to the feeling of weakness and of an inability to show courage. This example shows that, for the two men, the factual objective event did not have the same outcome or impact, because not only did the event not produce the same impressions but also because their inner history is different [66]. In short, the ‘decisive’ character of the event belongs less, in both cases, to the event itself than to the ‘reception’ that is attributed to it by a being, which at the same moment proclaims itself a ‘personal self’ [19].

We may note too that the event is also ‘constitutive’ of a ‘personal self’, that is, that it contributes to the development of an identity and a personal history. The event thus has a two-fold dynamic, it gives rise to an experience of an excessive nature, and it is, at the same time, constitutive (of a self) in the sense that it requires an adaptation or even a transformation to resolve the crisis. Our ‘self’ results from the addition of all the ‘traces’ left by our past experiences. To return in a rather simplistic way to our example, the doctor behaves as a “strong man”, whereas the young man behaves as a “weak man”. While the former experiences a creative transformation that renders him more robust and powerful, the latter was unable to engage in a similar transformation, and the event became a disaster, the memory of which will diminish little by little.

Tinnitus that appears and persists constitutes a significant event that will be remembered and can be dated: there was a before and an after the emergence of tinnitus. Tinnitus often appears suddenly and is immediately chronic and unpleasant, triggering a crisis in the subject. The first impression is always negative in that tinnitus is a sensory anomaly. The event (the presence of tinnitus) pursues its destiny in the existence of the subject according to the reception that it is afforded to it. Tinnitus constitutes an interesting clinical condition with regard to this point (reception of the event) in that its impact on life is very heterogeneous among subjects.

What is the imprint of tinnitus when it appears in the life of the subject, and what are the impressions with which it is associated? These impressions come well ahead of any negative automatic thoughts and/or erroneous beliefs and/or inappropriate survival behaviours, as suggested by the cognitive models [14]. It is also unlikely that these impressions might be linked to Pavlovian conditioning, as suggested by the model of Jastreboff [11]. What is the route traced by the tinnitus event in the existence of the subjects from the moment of its appearance and the original imprint it has left? What type of transformation does tinnitus trigger? This trajectory would appear to be linked as much to the subject’s personal history as to his or her disposition to be transformed by the event. The subject’s failure to be transformed to make room for tinnitus may destabilise the subject’s existence by causing closure and a withdrawal into oneself. As long as tinnitus is considered as an intruder, a disorder, an aggression, an injustice, or meaninglessness, tinnitus will keep the negative connotation of its initial impression.

### 12.4. Tinnitus and Plans

Human beings are unique, and their existence is lived within and outside themselves and outside the present time, taking root in the future and in plans. As was stressed by Porée, “*existence is an ecstasy; it is just a movement to get out of oneself*” [19]. This is also the sense of this quotation from Pascal: “*All the misfortune of humankind comes from one single thing, which is not knowing how to remain still in a room*” [67]. Nietzsche said something similar in this passage on humankind, whom he described as “the sick animal” *par excellence*: “*Assuredly they have dared more, innovated more, braved more, provoked destiny more than all the other animals together: they, the great experimenters who experiment on themselves, dissatisfied, insatiable, who battle for supreme power against the animal, nature and the gods–they, still untamed, the beings of the eternal future who finds no repose from their strength, pushed unceasingly by the burning spur that the future plunges into the flesh of the present: they, the bravest animal, the one with the richest blood, how could they not be exposed to the longest lasting and the most terrible of the diseases among those that afflict the animal?*” [32].

Tinnitus is a sensory anomaly that lasts. The impossibility of controlling tinnitus or getting control of it gives the subject a feeling of incapacity. The absurdity of its continuous presence may be upsetting. As we showed earlier, the spontaneously disagreeable and intrusive character of tinnitus depends in part on the characteristics of hearing, notably, its role as a warning system for the organism. The ‘physiological and psychological tension’ linked to the auditory system is such that any disruption to the harmony of the world of sound (tinnitus, hyperacousia, misophonia) is capable of impacting the subject in the minimal form of an increase in vigilance. Furthermore, it is of interest to note that the subjects presenting a severe loss of hearing seem less disturbed by tinnitus than subjects whose hearing is better preserved (personal observation). The auditory system, which has ceased to play its ‘radar’ role in these subjects, is more capable of integrating the background noise of tinnitus as a banal ‘body noise’ of the same order as muscle pain, for example.

Chronic tinnitus upsets the natural order of things in which existence habitually unfolds. The body, generally transparent and open to the world, becomes sombre and closes up. The relation to the world becomes difficult and turbulent. That is what Nietzche [45] meant in this extract cited in the book by Daniel Tammam [61]: “*But what does Man really know about himself... does not nature conceal most things even what concerns his own body in order to hold him prisoner of a proud and deceptive consciousness, at a distance from the folds of the intestines, at a distance from the body (…)*”. Tinnitus places the body in the forefront of consciousness, and it reminds us of itself as being a limit that cannot be gone beyond. By requisitioning the attention-related cognitive resources, to the extent of saturating them by its presence, hypervigilance vis-à-vis tinnitus abolishes all liberty and potentially any notion of a plan. Tinnitus thus blocks the life of the subjects: the subject is not drawn towards a fruitful future, and they lose their capacity for initiative. The decisive impact of tinnitus, which accounts for its weight on the quality of life, might be apprehended in terms of the reduction in the “power to live”. At one extreme, tinnitus does not impact the quality of life if it does not alter the subject’s life force, that is, if it does not impede the subject’s activities—profession, leisure activities, social life, rest, sleep, etc. On the other extreme, tinnitus is invalidating to an extreme degree if it prevents making plans and the ‘natural’ course of life. The subject is then in a state of maximum devastation, as is the case during episodes of unbearable pain (toothache, for example). The subject is reduced to suffering, that is, to an existence without plans and without temporality, where the unbearable present is renewed unceasingly.

### 12.5. Tinnitus, Temporality, and Suffering

If tinnitus as a sensory anomaly is a bodily excess that can bring life to a standstill, then suffering is an excess of the being, which calls into question the meaning of existence [44]. Suffering does not reside in the body, but it overshadows it; it is a turbulence in the meaning of existence. In this sense, suffering is not limited to sensations (pain or tinnitus) but may be experienced in ‘existential’ situations (bereavement or failure, for example). Suffering appears when time is confined in the present when plans cannot be projected into the future, and, in short, when the subject is a prisoner of a painful here and now. Suffering is the state that results from a crisis that lasts and from which there is no issue. The subject’s suffering thus depends as much on the tinnitus they are experiencing in the present, as on the tinnitus they anticipate the occurrence of in the future, which is a barrier to the future and to life. Humankind, that being of “the eternal future”, naturally “outside oneself”, leaning towards the horizon of tomorrow, open to the world, is thus exposed without meaning. It is the very machinery of life that is broken. The consciousness, which instinctively and mechanically seeks an issue to the crisis, is held in check. The whole situation is reminiscent of a scratched vinyl record, perpetually and purposelessly turning on its axis, which invariably sticks on the same groove, and which repeats incessantly the same sequence, without end. There is indeed a tomorrow in the life of someone affected by tinnitus, but a tomorrow that is only a repeat of the present. The tinnitus subject, like the chronic pain sufferer, is reminiscent of the torment of Sisyphus: a man (as a symbol of all humanity) condemned to the absurdity of existence by means of a rock that he attempts in vain to push up to the summit of a hill. What attitude can one have toward the repetition of meaninglessness? Tinnitus can wear down the subject to exhaustion and collapse (vulnerability, despair, and depression). Camus proposes a different path: to accept the futility of life, to embrace its absurdity, and to achieve acceptance of it. It is when we recognise and accept our lot that we are fulfilled and delivered: “*The struggle itself towards the summits is enough to fill the heart of man. We have to imagine Sisyphus happy*” [68].

The subjects who suffer question the aggression they are victims of. Why me, why does my body produce meaninglessness, why have I no more control over my body, and why am I unable to cope? (Marin and Zaccaï-Reiners, 2013) The loss of the natural order of things leads to an abyss, which plunges the subject into incomprehension, deprivation, and distress. The abyss of meaninglessness can then give the subject the illusion that they are predestined for ill and suffering, in a way that is arbitrary and unjust. We may finally accept suffering but on condition that we understand its source and its purpose. We can accept suffering for a good cause (freedom against oppression), for the just order of things (death from old age), for self-improvement, or even to experience the burning flame of life. On the other hand, suffering that is unjustified and void of meaning is odious and intolerable. This is what Nietzsche recalls in this passage: “*He [Man] suffers furthermore in many ways, he is above all a sickly animal: but his problem is not suffering itself, it is that he has no response to this anguished question: “Why suffer?”. Man, the most valiant, the most apt for suffering of all the animals, does not reject suffering in itself: he even seeks it, as long as he is shown its raison d’être, the reason for this suffering. The meaninglessness of pain, and not the pain itself, is the curse that up until the present weighs upon humanity* (…)” [32].

### 12.6. Tinnitus and Sociology

The last point that I wish to develop here is the possible relationship between tinnitus, the way it is experienced, and how it can impact the subject and social organisation. Is there a link between these two aspects that may seem at first sight to be quite far apart? The social organisation changes in the function of the place (the society in question) and of history, which leads us to wonder whether tinnitus is the same ailment throughout the world and throughout history. In other words, does tinnitus have the same status of ‘disorder’ as defined in contemporary Western countries (Europe and its extensions in the Americas, etc.) as in societies that are fairly distant from this model and are less developed? In addition, was the level of suffering from tinnitus the same in the Middle Ages as what is reported today? If the answer to both questions is no (variation in function of the type of society and of history), then social factors play a role in “the appearing” of tinnitus. In other words, is the tolerance of tinnitus, which was described early in history (from the Mesopotamian civilisation), influenced by the major trends in society? The aim of this paragraph is to show the influence of social processes in the experience of tinnitus and in what ways they may play a role in the whole lived experience.

#### 12.6.1. The Civilisation of Manners and Tolerance to Tinnitus

Norbert Elias described how the organisation of society gradually changed over the course of history [69,70]. In short, with pacification between individuals, people became more and more interdependent. This interdependence was accompanied by social practices, invented in the laboratory of courtly society, and strictly regulating behaviour (courtesy). Ill-mannered but tolerated behaviours (spitting at the table, wiping one’s nose on one’s clothing, clearing one’s throat, etc.) were gradually prohibited and considered as repugnant and taboo. These behaviours had to be severely repressed by conditioning during the critical period of childhood and adolescence. The purpose of this ‘training’ was firstly, to avoid bothering others and perhaps to inform them of one’s social rank. The individual had to interiorise the rules and hide their emotions and impulses. Therefore, this process was achieved with the development of an interiority. This ‘inner self’ is the private refuge, that of security and secret speculation. It is probable that today’s ‘inner self, in our individualist and socially distant societies, is hypertrophic compared to that of medieval people. Yet, the ‘inner self’, which differentiates an inside and an outside, also defines an assault. There is no assault if there is no protective barrier. The more the ‘inner self’ is developed and deep-seated, the more it offers protective asylum, and the more any intrusion may be perceived as forced entry. In the context of our society of ‘control’, where behaviour is supposed to be strictly mastered, and where we shelter in an inviolable private space, the least intrusion may be experienced as aggression. The development of social rules that apply to oneself are matched by heightened expectations towards others, and, beyond that, towards the world and diminished tolerance of intrusion. If tinnitus may be considered as sensory background noise, unpleasant but finally trivial, then its ‘appearing’ is quite different if it disturbs the ultimate refuge of the subject. To put it more crudely and rather simplistically, medieval people who lived in a house with a single room, with all their family and sometimes animals, did not have the same ‘inner self’ nor the same relation with the auditory world as people today. In that society, where the separation between people was much less clear than today, in this noisier and coarser life, it is likely that tinnitus did not have the same incidence as it has today. One might add that medieval people did not have any choice regarding their health either. They had to put up with all their ‘sensory background noises’, which were part of normal life. For people today, ‘silence is golden’, and it has become a luxury and a social distinction. As Pascal Quignard said, “*Silence has become today’s vertigo. Its ecstasy*” [56]. Silence is a social marker: the leisured classes with their private houses set in an extensive space dispose of a preserved sound environment, free of pollution from any close neighbours.

We dispose of little data on the ‘social’ aspects of tinnitus, but they are on the other hand plentiful concerning pain. René Leriche, a surgeon at the front during the First World War, documented the different ways, a priori cultural, of reacting to pain [71]. It is possible that if the threshold for sensitivity appears to be identical for all human societies, then the threshold for tolerance of pain seems to depend on social and cultural organisation [72]. Within the same society, differences in attitude with regard to pain may be spectacular [17]. In working-class communities, people often put up with pain as long as it is not too intense. Toughness is valorised: one has to be ‘tough as nails’ ‘to not mind’. Life calls for a certain toughness: ‘being too soft on yourself’ is no help in earning your living. This toughness towards hardship is a source of pride in working-class communities; it is shared and is held up as a virtue. In rural society, toughness towards pain is also very marked [17]. The workload and the modest income small-scale farmers earn give little opportunity for dwelling on a backache or a persistent cold. As concerns the middle classes and, above all, the well-off, the attitude vis-à-vis the impression of pain is very different: “*Any pain is treated as soon as it appears. The ailment barely has time to set in. As soon as an unusual and lasting sensation gets beyond the threshold of consciousness, a consultation with a doctor is compulsory. The attention paid to morbid affections shows a threshold that is distinctly lower than is the case for the other social strata*” [17].

#### 12.6.2. Tolerance and “Pain Killers”

It is also stressed that the widespread use of painkillers has greatly contributed to lowering the threshold for tolerance to pain. If we make a parallel between pain and tinnitus, it is probable that these attitudes relative to ‘pain’ may be transposed to tinnitus, which might be considered as a kind of pain or an ‘impression of pain’. The threshold of tolerance for tinnitus is therefore also probably lower in our industrialised societies of today. What was previously held up as a virtue is no longer: to be subjected to pain has become absurd and intolerable, which may paradoxically contribute to making the symptom really intolerable. It is rather as if the suffering were increased by the simple lack of acceptance of suffering: “*The polls show that the fear of suffering induces a distinctly higher degree of terror than the fact of dying. Pain is absolutely meaningless, pure torture*” [17]. In a society where treatment for pain is considered a right, the presence of tinnitus is unacceptable. And yet, as Louis Lavelle wrote (cited by Le Breton, 2012): “*Each one of us probably only dreams of rejecting pain as soon as it hits us; but when one thinks back over one’s life, then one sees that it is the pain that has been endured that has had the greatest effect; it has marked us: it has given life its seriousness and its profundity; it is also from it that one has learned the most essential lessons regarding the world one is called to live in and the meaning of its destiny*” [73].

#### 12.6.3. The Society of the “Self-Made Man”

Furthermore, our society shifted from a society of discipline to a society of individuals in about the 1960s [15]. A society of the individual promotes responsibility, independence, individual initiative, and self-control. The priority is to become oneself: “*Everything happens as if, in our civilisation, every individual has their own personality as a totem*” wrote Levi-Strauss [74]. The aim, in addition to being oneself, is to be happy. Yet, the liberation of the individual has been achieved at the cost of insecurity regarding one’s identity, an instability of self-respect. The success of the subject is now incumbent on the subject themselves and less on their social origins. The subject thus bears the responsibility for their failures or, rather, their lack of success in the eyes of society. Through the feeling of incapacity, inadequacy and exhaustion, in this society that valorises the ‘self-made man’ and the grand lifestyle, depression (or the famous ‘burn-out’) is a possible realisation of the distress associated with the failure to meet the social norms: “*I do not know if people were more inhibited previously than today, but inhibition is something that is of course much more visible and more of a handicap in a society that calls more for initiative and the capacity for action than for docility* (…)” [75]. In this context, where individuals seek valorisation, tinnitus may be seen as an additional difficulty, as an impediment to self-fulfilment, and may provoke a strong reaction and a violent rejection.

#### 12.6.4. Tinnitus: A Condition in the Periphery of Scientific Medicine

Finally, scientific medicine is becoming increasingly reductionist and leaves on the periphery what one might call malaise or feelings of insecurity in society [15,76]. The more scientific medicine becomes, the more its range of application is extended, and paradoxically, it leaves on the sidelines disorders that become increasingly severe the more they are marginalised. The subjects are often astonished to find that not only does traditional medicine have nothing to offer them but it no longer even knows how to talk to them or listen to them: “I’m suffering and you don’t understand me” [76]. So, the feeling of malaise is treated with a range of ‘alternative’ approaches (on the periphery of the increasingly engineering-style approach in medicine, from sophrology and hypnosis to positive psychology and CBT).

## 13. Discussion

I offered a description of tinnitus based on a framework of interpretation inspired by the phenomenological approach. Using this approach, tinnitus presents many similarities with chronic pain. The doctor, René Leriche, provided well-known reflections on pain, and in particular its frequent futility: “(…) *Pain is only a contingent symptom; tedious, noisy, unpleasant, often difficult to get rid of, but which usually is of no great value, either for diagnosis, or for prognosis*” [17,71] This description could well apply to tinnitus! It is remarkable that pain should be compared to noise using the adjective ‘noisy’! He thus defines health as “*life in the silence of the organs*”. There is indeed in sound, in its deafening, rumbling, piercing, vociferous, howling, shrill character, the expression of pain that is its raging energy. He adds: “*The number of diseases that it reveals is tiny, and often when it accompanies them, it only serves to deceive us. On the other hand, in a few chronic conditions, it appears to be the whole disease, which without it, would not exist.*” Here, as for tinnitus, the disagreeable and lasting becomes in itself the cause of the symptom of an ailment or of a ‘disease’. The nature of the symptom changes, it is at once a symptom and a cause.

Tinnitus comes within the register of feeling and not of perceiving. Tinnitus is given as it is, without intention or distance, and without exploration of the world or any attempt to objectivise it. Tinnitus is not a perception in the sense that it tells us nothing about the world. If transitory tinnitus is considered ordinary background noise, then tinnitus that lasts appears clearly as turbulence that fills the consciousness. As with chronic pain, tinnitus gives a certain tonality to the world: it infiltrates all the activities of the consciousness, from sensations to judgements. It is the unreachable horizon, and in this sense, it causes separation not only from the world but also from others. Chronic tinnitus is experienced as an event, which defines a before and an after its emergence. Tinnitus may be compared to forced entry, which is all the more serious inasmuch as people today have developed a hypertrophic ‘inner self’, and thus a hypersensitivity (or a diminution of tolerance), to intrusion. In the face of this aggression and this invasion, the reaction of the subject is one of defence, resistance, and, therefore, of heightened vigilance vis-à-vis the tinnitus-aggressor. Tinnitus then triggers a closing off from the world and the cutting back of plans. This condition prevents the normal course of time, it traps the subject in the unceasing repetition of a painful present and prevents the deployment of the future. The life of the subject is hunkered down in a present that repeats itself endlessly like a scratched record. Consciousness, which instinctively and mechanically seeks a way out of the crisis, is at a loss. It is the very machinery of life that is broken down, its most profound mechanism, that of its opening to the world. The absence of any issue from this state of pain leads to suffering, that is, a deconstruction of the subject themselves and the disorganisation of their existence, which is existential (“Why me, why suffer?”, etc.). Suffering may wear down the subject’s capacity for resistance to the point of exhaustion and collapse.

To my knowledge, there is at present no tool for assessing the impacts of tinnitus according to the frame of reference I just elaborated. The scores of psychometric assessments, like questionnaires and VAS tools, define the ailment as if it were a linear variable: the higher the score, the greater the discomfort. Yet, we feel that the severity of tinnitus is fundamentally non-linear, that is it manifests itself in categories of severity. Suffering is the most heterogeneous and the most varied category since it is also the most personal. Suffering can range from exhaustion to existential questioning via feelings of vulnerability and incapacity. It is of fundamental importance to properly define the way tinnitus is accepted in order to better adapt the treatment of it.

### 13.1. Acute vs. Chronic Tinnitus

I also feel that the frame of reference proposed in this article may provide matter for the debate on acute vs. chronic tinnitus, notably, the issue of when to consider that tinnitus is a chronic clinical condition [2]. I suggest that the transition from acute to chronic tinnitus needs to be considered according to two separate approaches: one is pathophysiology and the other is the lived experience. The dynamics of pathophysiology and the lived experience are likely different.

From the physiological point of view, it is difficult to define a duration beyond which acute tinnitus becomes chronic. If tinnitus was triggered by an auditory trauma, then we might invoke the impact of the trauma on the cochlea and, above all, the possibility of recovery from this impact. How long after the auditory trauma can we say the damage to the cochlea is definitive? One might suppose that not much is happening (at the peripheral level) one month after an auditory trauma and that this lapse of time seems reasonable as a limit beyond which the lesions to the cochlea can no longer recover [77,78]. Tinnitus is also linked to the impact of auditory trauma on the auditory centres, which are in fine implicated in the perception of tinnitus. The plastic alterations to the auditory centres, which might be linked to tinnitus, develop relatively rapidly, and take from a few minutes to a few days (depending on the mechanism considered) to be established [79,80]. In conclusion, based on the physiopathological mechanisms of tinnitus induced after an auditory trauma, we might estimate that tinnitus becomes chronic a few months after its emergence [2,81,82]. However, the transition from acute to chronic tinnitus depends heavily on the tinnitus sub-type and pathophysiology (i.e., the stability of the tinnitus mechanisms).

As regards the lived experience of tinnitus, it is much more difficult to determine a clear borderline between acute tinnitus and chronic tinnitus. We saw earlier how the acceptance of an event is a process specific to the subject and how the outcome of the event is both highly variable and unique. For this reason, it seems unreasonable to fix a single time frame for all tinnitus subjects. If we consider suffering as the consequence of an impression of pain that lasts, then we may consider that the suffering is similar and comparable to many painful events in life (sickness, physical pain, moral pain, bereavement, failure, etc.). DSM V broke new ground by fixing a very short time frame (two weeks) beyond which an unbearable sense of bereavement might be categorised as resulting from a psychological disorder. Not only is the diagnosis of the disorder in question left to the subjective assessment of the psychiatrist, but it is not at all certain that such a pathologisation of bereavement beyond a certain time would be of benefit for the subject, while there is also a real risk of committing an error of diagnosis. I feel, contrary to the idea that it is necessary to pathologise a clinical condition to treat it, that it is absolutely normal to live through a painful and perhaps long-lasting experience when the event undergone is particularly traumatic. Furthermore, I wonder what is the advantage of imposing such a norm.

One’s existence is torn by a crisis and must be reconfigured if it is to again follow its course as well as it can. Certain subjects are devastated by the emergence of tinnitus, whereas others, on the contrary, accept its appearance more easily. Each subject has his or her own way of integrating the pain or the event. In other words, tinnitus implants a unique outcome in each subject. In particular, tinnitus has a more or less profound impact on the subject: a power of deconstruction that is more or less strong. Here, I suggested that the reception of tinnitus is certainly related to historical, psychological, and social factors.

In any case, it is clear that the suffering experienced should be described in a complete and precise manner. In my view, tinnitus becomes chronic and thus significant from the clinical point of view as soon as it gives rise to questioning, increased vigilance, and a certain level of discomfort. In a way, the subject who comes to consult for tinnitus does so, by definition, for chronic tinnitus (which is the only kind to constitute an event); the fact of their complaint alone defines chronic tinnitus. Acute or transitory tinnitus is, by definition, tinnitus that does not constitute an event, which is not associated with a crisis. Once tinnitus is chronic, it may be considered as ‘in progress’ as its severity is not stable in time. When tinnitus has acquired the status of sensory anomaly giving rise to a crisis in the subject, it may have devastating effects on him or her, drastically reducing the quality of life. The severity of tinnitus may increase rapidly with time, over the course of consultations, and, above all, failures, frustrations, and clinical feedback. It is essential that the subject be treated as quickly as possible to avoid the development of negative thoughts, which might add to the sensory anomaly and thus aggravate the discomfort. I extensively presented in this article the point of view that tinnitus is a sensory background noise, a disorder that may cause a failing of consciousness, well before the emergence of ‘negative thoughts’, as suggested by the cognitive models. That being said, the discourse of the clinical practitioners or picked up from social media, such as ‘there is nothing to be done’, may give rise to negative thoughts. This discourse describes the psychoacoustic characteristics of tinnitus, but not at all the full scope of its phenomenology, that is, its ‘appearing’, i.e., the lived experience. But the ‘appearing’ of tinnitus may be radically altered. The impact of tinnitus may be greatly reduced if the subject is reassured by the discourse of the specialist consulted. Sometimes, the subject only needs some calming feedback from a specialist to consider tinnitus as a minor sensory background noise like an old backache, which is moderate and manageable [13]. By reducing the anxiety-generating uncertainty related to tinnitus, the information provided on the symptom is among the things that can most help tinnitus subjects to make their experience of it quite insignificant and finally acceptable [83].

After a certain amount of time, of unknown duration and certainly variable in each subject, the severity of tinnitus is stabilised around a certain level. If tinnitus is a source of discomfort for the subject, then I would call this ‘poorly tolerated tinnitus’. The severity of tinnitus may be stabilised at very different levels, ranging from light (minor impact) to very severe (the consciousness is saturated and life is impeded). Finally, the subject can take control of his or her tinnitus and envisage a way out of the ‘crisis’. Each subject will have their own way of getting out of the ‘crisis’, depending on the level of suffering and deconstruction that their tinnitus has caused. I would call this stage ‘chronic tinnitus in remission’. The final stage is then ‘stable and well tolerated’ tinnitus, which no longer causes discomfort in the subject or the stage of ‘total remission’ (complete disappearance of tinnitus).

### 13.2. Estimation of the Impact of Tinnitus

Studies on the complete outcome of tinnitus from its emergence are lacking. It would be interesting to have a more complete overview of the duration of the various stages, their inter-individual variability, and the proportion of subjects reaching the stage of ‘stable, well tolerated’ or ‘total remission’. For that, it is necessary to possess suitable tools for assessing the severity of tinnitus and its real impact on the lives of the subjects. We saw earlier that it would no doubt be useless to seek a convincing cause of the impact of tinnitus in psychoacoustic variables (in particular, the loudness).

There remain questionnaires that address the impact of tinnitus by dividing up the lived experience of the subjects (anxiety, concentration, anger, depression, etc.) [84,85]. The problem with the questionnaires is that they itemise emotions that are difficult to estimate separately inasmuch as they are experienced in an integrated and interdependent way [76]. Given that the severity of tinnitus is proportional to the reduction in the power to live (life is blocked, plans are reduced, and time is abolished) that it causes, I feel it would be interesting to develop semi-directed interviews to allow subjects to describe in real terms how tinnitus has disturbed the course of their life. If tinnitus alters the cognitive capacity, in particular, attention-related resources (the power of concentration) (by saturating the consciousness), then the subject will complain of not being able to perform a certain activity rather than of a problem of concentration. It would therefore be of interest to question subjects on the activities they have had to give up or that were made very difficult by tinnitus, if the subjects still have plans, what is, in their view, the degree of invalidity caused by tinnitus, etc. The problem of tinnitus, as chronic pain, or any other disorder, is that it diminishes people. Someone who has just announced a complaint feels diminished. Below is a non-exhaustive list of questions to ask the subjects:How would you estimate the degree of invalidity caused by tinnitus?At what moments in your life is tinnitus a cause of impediment?In what way is tinnitus a cause of impediment?Which activities have you given up, or have become less agreeable, because of tinnitus?Which activities do you still do despite tinnitus?Has having tinnitus altered your plans? If so, which?How do you see the future?Do you have any plans?

The whole difficulty is to quantify an overall incapacity (and/or decline in the quality of life) that can be attributed to tinnitus. It is urgent to undertake research of this type, taking inspiration from the field of pain, in general, and that of fibromyalgia, in particular.

### 13.3. Psychopathology and the DSM

Recently, it was suggested to define the impact of tinnitus on the lives of subjects as a somatoform disorder, which is a category of disorder newly created in the DSM [2]. Pathologising tinnitus distress, in the sense of a category in the DSM, can only be performed by defining a norm in terms of the severity of the disorder and its duration. This issue is difficult with regard to suffering as it directly raises the question of the possibility of an existential norm. What is the moral pain that must not be exceeded after a bereavement, separation, chronic pain, or tinnitus? What is the maximum period of suffering after a bereavement, separation, or tinnitus? Is it possible to define an existential norm? Who is capable of defining such a norm (body of specialists) or of applying it (the specialist in consultation)? In the field of health, the concept of a norm should be treated with caution inasmuch as nature can constantly reinvent new life norms. One can live through an illness and its symptoms without feeling ill. To feel ill is to feel diminished [63].

The existential crisis that arises for the subject from a disturbance in the course of their life is shared by all, and it is precisely this that makes us human. But this crisis is also unique and personal inasmuch as it is experienced in the first person by a subject who has their own history. The DSM meets a real need to categorise mental disorders. I feel the problem is different when it is a matter of the suffering of a tinnitus subject since medical and biological thinking impinges on the territory of values and subjectivity. Is it not the same thing to define disorders related to autism, schizophrenia, or depression (the task is already an arduous one!) as it is to define the severity of tinnitus according to a psychopathological frame of reference? Does the subject with tinnitus present symptoms of depression? So be it. One might wonder to what degree tinnitus triggers these symptoms. Were they latent but not ‘expressed’? What was the source of the symptoms of depression in the subject’s history and perhaps biology? It may well be envisaged that tinnitus takes over the outcome of depression in certain subjects, in other words, it is the triggering factor. There is no need, however, to create an additional ‘psychiatric’ entity for that. Much more is required from the classification of the impact of tinnitus in the DSM than it can offer. Not only does it not offer anything new in terms of our understanding of the psychopathology of tinnitus, but, furthermore, it is likely to pathologise uselessly and dangerously the subjects’ suffering. What could be the impact of such pathologising on a subject who is suffering? As long as the treatment of the subjects is not improved, it can only be negative: “You are suffering because you are unable to reconfigure yourself in such a way as to be able to integrate the tinnitus in your existence. *This lack of adaptability is pathological*. There are approaches for attempting to give you the means to adapt to this situation”. There is a risk of making the subject feel guilty, especially in an age when society attaches such importance to capability and autonomy [75]. Medicine has no need to classify suffering in the DSM to try to help the subject. In my mind, this classification raises more problems than it solves, the first of them being the definition of a norm. Furthermore, the classification of a disorder in the DSM places the disorder under the dominion of psychiatry, a discipline that, like the others, has its own bias. I plead for prudent reflection, open and multidisciplinary, integrating psychology but also other fields in the humanities, notably, philosophy, history, and sociology [86]. A broad-based reflection is the only way to achieve a portrait of tinnitus.

### 13.4. Tinnitus Management: Towards an Existential Metamorphosis

Finally, despite their despondency, most subjects suffering from tinnitus attempt to get over it. Their action then takes the form of a complaint (Ricœur, 2013). A suffering subject is free to choose between two attitudes: “*They can deny their distress and consider the situation as a disorder which must be borne with courage and serenity. Or they can accept the distress at the cost of an existential metamorphosis which will make them reconsider their identity and the meaning of life*” [33]. Having no vital significance, the suffering must be resolved in an existential sense. If the state of pain that is the source of the suffering is chronic, then it is possible to demolish the suffering at the cost of metamorphosis. It is destroyed within the being themselves, in their manner of considering the pain, and, more generally, in the core of their existence.

The emotional response, which is a natural defence response, should progressively give way to transformation. Life must find another way to continue to follow its course. The subject should reconstruct their life taking into account the new condition. But that takes time, willpower, and a favourable and supportive environment. It is important to recognise the painful moments in life and not to pathologise this period of reconstruction. We cannot determine in advance what the subject will need during this period of distress, nor how long it will last. The subject should be accompanied, encouraged, and supported, but reconstruction is an adventure that must take place within the subject themselves. The existential reconfiguration, like existence, cannot be delegated to anyone: life must be lived.

Tinnitus is disturbing for only 20% of subjects, and many of them get better with time [87]. Certain approaches, such as cognitive and behavioural therapy (CBT), for example, are recognised as improving subjects’ quality of life [88,89,90,91]. They help subjects control their emotions, avoid counter-productive behaviour, and take an objective view of their tinnitus. They encourage subjects to accept their tinnitus and to put it in the background, as one would with an external sound nuisance or a minor sensorial background noise. CBT is useful to reduce the brazier of suffering, but it is, on the other hand, inoperative to address the meaning of life and profoundly transform the being. This work of metamorphosis is achieved by the subject themselves with the possible support of a physician.

The psychopathological treatment should include a minimum of elements. The subject goes to consult to feel better and at least to be understood when they register their complaint. Tinnitus is less ‘heavy’ to bear when the subject is not left alone to await the return of better days. The physician thus gives the subject the strength to enable them to cope with tinnitus by themselves. They should encourage talking and not give up. One might say that the primary function of the physician is a Socratic mission, in other words, to help the subject to ‘deliver’ their complaint. The subject who comes for a consultation is in a state of mind to find a solution to the crisis that will enable life to follow its course again towards the future. It must be understood that the subject who is asking for care feels the deepest helplessness, the absurdity of existence: “*The absurd is born of this confrontation between the human appeal and the unconscionable silence of the world*” [68]. In this context, the ill-chosen utterance that should not be pronounced is “I can do nothing for you, you’ll have to learn to live with your tinnitus (…)”, which crushes the subject and condemns them to helplessness and despair. On the contrary, the physician should do everything they can to give the subject new hope and a time frame that is a future with future plans: “Once the storm is over, which we will get through together, you will make new plans and your life will go on again, and the tinnitus will no longer be an obstacle: at the worst, it will be a minor background noise, like your usual lumbar pains, etc.”.

The sooner the patient is treated, the quicker the processes that cause suffering and possible exhaustion and collapse are neutralised, and the sooner the subject can start on the road towards the end of the crisis and tinnitus referred to as ‘in remission’. On the other hand, it is important that the subject should have reasonable expectations concerning his or her tinnitus, that is, which may satisfy the subject without leading to frustration. As I mentioned earlier, consciousness is ‘drawn’ towards the future and seeks a solution that will offer an end to a crisis. But the most reassuring and immediate solution would be the suppression of tinnitus or the drastic reduction of its intensity. This goal is not always achieved, and care must be taken so that the subject is not disappointed and does not immediately close up again after opening up to treatment. One may hope for the suppression of tinnitus, but we must start by considering tinnitus as a normal background noise.

## Data Availability

Not applicable.

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
