# Peer review of "The Analogy between Tinnitus and Chronic Pain: A Phenomenological Approach"

_brainsci, 2023, doi:10.3390/brainsci13081129_

Round 1
Reviewer 1 Report
Dear Author,
I have reviewed the manuscript. My response is given in a point-by-point manner below.
Sincerely,
Analogy between tinnitus and pain: a phenomenological approach
Title:
· The title is so good and concise, but based on my search I found the article with the this title
“The analogy between tinnitus and pain: a suggestion for a physiological basis of chronic tinnitus (PMID: 2820913)”. For this, I suggest you to change little the title to attract more the audience.
Abstract:
· In 3rd line, for better of expression, It’s better to remove the “with which it may be associated”.
· In 5th and 7th lined, it is better to use “paper” instead of “article”.
· In 4th line, I suggest you to add the “psychological factors” in the text, to direct better the attention of the audience.
· I think for this valuable review article, add the “keyword section” to the text.
Main Text:
1. Introduction
· In 4th paragraph, the “timbre” is not suitable. I suggest to remove it.
· In 5th paragraph, it is better to use “paper” instead of “article”.
2. Critique of behavioural and cognitive psychology
2.1 The behaviourist model:
· I think is better to use “behaviorism” instead of “behaviourist”.
2.2. The cognitivist model
· I suggest you to apply “cognitive model” instead of “cognitivist model”.
· In 12th line, I suggest to use “Sound therapy techniques” or “Masker devices” instead of “using acoustic devices”.
· In 2nd paragraph, what’s the means of “(…)”?
2.3. Some reflections on the behaviourist and cognitivist models
· This part is comprehensive, I just suggest you to write it briefly.
3. What is phenomenology, 'the science of phenomena'?:
· The topic "phenomenology" was very good for audience like me who may not be very familiar with its concept.
4. Sketching out a phenomenology of tinnitus?
· Based on my opinion, it is comprehensive and flawless.
5. What is perceiving?
· This section is excellent, but I suggest you to present it more briefly. For example, some explanation of 2nd, 3rd and 5th paragraph is redundant.
6. Body, affectivity, emotion
· This section seems complete.
· I suggest you to blend 5th and 6th section with together.
7. What is pain?
· I strongly suggest blending 7th and 8th sections and writing it briefly.
8. The affective part of pain
· I strongly suggest blending 7th and 8th sections.
9. From pain to suffering (distress)
· It may be better to apply “distress” in the title of this section, instead of “suffering”.
· I think other parts of this section are flawless.
10. From pain to 'impressions of pain'
· In 2nd paragraph, applying “hearing” instead of “audition” is better.
11. What is hearing?
11.1. Hearing is the channel of vigilance
· At the end of 1st paragraph, the “localization” may be better than “localizing” and “localize”.
· Other parts are comprehensive.
11.2. Controlling the auditory system
· It was so comprehensive, especially the examples, which directing the attention of readers to the tinnitus.
12. What is tinnitus?
12.1. Tinnitus is of the order of 'feeling'
· In 1st paragraph, I suggest you to remove the “timbre”.
12.2. The implications of chronic tinnitus
· In 1st paragraph, it may be better to use “somatic” or “physical” instead of “corporeal”.
12.3. Tinnitus and event
· This section is very long, I recommend making it more concise.
12.4. Tinnitus and plans
· In 1st line, it’s better to write “Human beings are unique” instead of “Humans are unique animals”.
12.5. Tinnitus, temporality and suffering.
· In 1st paragraph, I think “tinnitus subject” is better than “tinnitus sufferer”.
· From a writing point of view, at the end of paragraph, the type of font seems incorrect.
12.6. Tinnitus, individuals and society
· This section is comprehensive, but its length can make the audience tired. Therefore, I recommend removing its redundant sentences.
13. Discussion
· At the end of 2nd paragraph, the type of font seems difference.
· In 3rd paragraph, I suggest you to write “The score of psychometric assessments, like questionnaires and VAS tool” instead of “The scores of the questionnaires”.
13.1. Acute vs. chronic tinnitus
· Based on previous papers, “Chronic tinnitus” has duration at least 3 and or 6 months. I think one month is incorrect. Please revise it at 2nd paragraph.
13.2. Estimation of the impact of tinnitus
· In 3rd paragraph, the bullets have a different style and size. Editing it may be suitable.
13.3. Psychopathology and the DSM
· In 1st paragraph, it’s better to apply “Tinnitus patients” instead of “Pathologising suffering‘.
13.4. Towards treatment
· I recommend that other treatment options for tinnitus such as sound therapy or neuromodulation techniques (such as tDCS and rTMS) are added, so not to create a bias for the audiences that CBT is the only effective treatment option for tinnitus.
Minor editing of English language required.
Author Response
Reviewer 1
I would like to thank this reviewer for his thorough review of the article and his comments.
Title:
- The title is so good and concise, but based on my search I found the article with the this title
“The analogy between tinnitus and pain: a suggestion for a physiological basis of chronic tinnitus (PMID: 2820913)”. For this, I suggest you to change little the title to attract more the audience.
I changed the title to:
“The analogy between tinnitus and chronic pain: A phenomenological approach”
Abstract:
- In 3rd line, for better of expression, It’s better to remove the “with which it may be associated”.
Corrected as suggested.
- In 5th and 7th lined, it is better to use “paper” instead of “article”.
The manuscript has been corrected by a professional (English) translator who chose to use “article”. I'll stick to “article” throughout.
- In 4th line, I suggest you to add the “psychological factors” in the text, to direct better the attention of the audience.
Corrected as suggested.
- I think for this valuable review article, add the “keyword section” to the text.
Keywords have been added
Main Text:
- Introduction
- In 4th paragraph, the “timbre” is not suitable. I suggest to remove it.
Timbre can be defined as what is not loudness or pitch so I decided to leave it.
- In 5th paragraph, it is better to use “paper” instead of “article”.
See above
- Critique of behavioural and cognitive psychology
2.1 The behaviourist model:
- I think is better to use “behaviorism” instead of “behaviourist”.
Corrected.
2.2. The cognitivist model
- I suggest you to apply “cognitive model” instead of “cognitivist model”.
Corrected.
- In 12th line, I suggest to use “Sound therapy techniques” or “Masker devices” instead of “using acoustic devices”.
Corrected.
- In 2nd paragraph, what’s the means of “(…)”?
Part of McKenna’s paper is quoted.
2.3. Some reflections on the behaviourist and cognitivist models
- This part is comprehensive, I just suggest you to write it briefly.
This section has been shortened.
- What is phenomenology, 'the science of phenomena'?:
- The topic "phenomenology" was very good for audience like me who may not be very familiar with its concept.
Thanks.
- Sketching out a phenomenology of tinnitus?
- Based on my opinion, it is comprehensive and flawless.
Thanks.
- What is perceiving?
- This section is excellent, but I suggest you to present it more briefly. For example, some explanation of 2nd, 3rd and 5th paragraph is redundant.
This section has been shortened.
- Body, affectivity, emotion
- This section seems complete.
- I suggest you to blend 5th and 6th section with together.
Ok
- What is pain?
- I strongly suggest blending 7th and 8th sections and writing it briefly.
This section has been reorganized.
- The affective part of pain
- I strongly suggest blending 7th and 8th sections.
Ok
- From pain to suffering (distress)
- It may be better to apply “distress” in the title of this section, instead of “suffering”.
- I think other parts of this section are flawless.
Ok
- From pain to 'impressions of pain'
- In 2nd paragraph, applying “hearing” instead of “audition” is better.
Ok
- What is hearing?
11.1. Hearing is the channel of vigilance
- At the end of 1st paragraph, the “localization” may be better than “localizing” and “localize”.
- Other parts are comprehensive.
Ok
11.2. Controlling the auditory system
- It was so comprehensive, especially the examples, which directing the attention of readers to the tinnitus.
Ok
- What is tinnitus?
Ok
12.1. Tinnitus is of the order of 'feeling'
- In 1st paragraph, I suggest you to remove the “timbre”.
We decided to leave “timbre”. See above.
12.2. The implications of chronic tinnitus
- In 1st paragraph, it may be better to use “somatic” or “physical” instead of “corporeal”.
Ok
12.3. Tinnitus and event
- This section is very long, I recommend making it more concise.
This section has been shortened.
12.4. Tinnitus and plans
- In 1st line, it’s better to write “Human beings are unique” instead of “Humans are unique animals”.
Ok
12.5. Tinnitus, temporality and suffering.
- In 1st paragraph, I think “tinnitus subject” is better than “tinnitus sufferer”.
- From a writing point of view, at the end of paragraph, the type of font seems incorrect.
Ok
The font is ok.
12.6. Tinnitus, individuals and society
- This section is comprehensive, but its length can make the audience tired. Therefore, I recommend removing its redundant sentences.
This section has been reorganized (divided into sections) and shortened.
- Discussion
- At the end of 2nd paragraph, the type of font seems difference.
No it is not.
- In 3rd paragraph, I suggest you to write “The score of psychometric assessments, like questionnaires and VAS tool” instead of “The scores of the questionnaires”.
Ok
13.1. Acute vs. chronic tinnitus
- Based on previous papers, “Chronic tinnitus” has duration at least 3 and or 6 months. I think one month is incorrect. Please revise it at 2nd paragraph.
Ok
13.2. Estimation of the impact of tinnitus
- In 3rd paragraph, the bullets have a different style and size. Editing it may be suitable.
No they haven’t.
13.3. Psychopathology and the DSM
- In 1st paragraph, it’s better to apply “Tinnitus patients” instead of “Pathologising suffering‘.
“Pathologizing suffering” has been replaced by “Pathologizing tinnitus distress”.
13.4. Towards treatment
- I recommend that other treatment options for tinnitus such as sound therapy or neuromodulation techniques (such as tDCS and rTMS) are added, so not to create a bias for the audiences that CBT is the only effective treatment option for tinnitus.
The point here is not to list the various treatments: the point is that tinnitus must be integrated by tinnitus patients at the cost of an existential metamorphosis. We changed the section’s title to:
“Tinnitus management: towards an existential metamorphosis”.
And a sentence has been slightly changed:
“CBT is useful to reduce the brazier of suffering, but it is on the other hand inoperative to address the meaning of life and profoundly transform the being.”

Reviewer 2 Report
The analogy between tinnitus and pain is well reported in the literature. The author proposes in this article a innovative phenomenological approach to tinnitus also towards effective treatments. The author carries out a complete and correct study not only on tinnitus but on all phenomena of perception.
Why the author speaks of "Analogy between tinnitus and pain: a phenomenological approach" in the title and in the text (correctly) of chronic pain?
Pain is a symptom, chronic pain is a disease. Moreover the similarities between tinnitus and chronic pain involve the frontolimbic striatal pathways that I don't find mentioned.
I have also some minor suggestions.
Introduction:
- the reference of the work by the AFREPA is not reported.
-the definition of tinnitus disorder (De Ridder et al. 2021) and auditory allucinosis (Messina et al. 2022) could be added.
Pag.26: the temporal definition of acute and chronic tinnitus (> 3 months for some authors, > 6 months for other authors) should be added with references and discussed.
The manuscript is written in fine English and requires a minor text editing.
Author Response
Reviewer 2
The analogy between tinnitus and pain is well reported in the literature. The author proposes in this article a innovative phenomenological approach to tinnitus also towards effective treatments. The author carries out a complete and correct study not only on tinnitus but on all phenomena of perception.
I would like to thank this reviewer for his valuable comments.
Why the author speaks of "Analogy between tinnitus and pain: a phenomenological approach" in the title and in the text (correctly) of chronic pain?
Pain is a symptom, chronic pain is a disease. Moreover the similarities between tinnitus and chronic pain involve the frontolimbic striatal pathways that I don't find mentioned.
The title has been changed to :
“The analogy between tinnitus and chronic pain: A phenomenological approach”
Regarding the reviewer’s comment about the “frontolimbic striatal pathways”: the article does not deal with the mechanisms of tinnitus or pain but with the way they are experienced.
I have also some minor suggestions.
Introduction:
- the reference of the work by the AFREPA is not reported.
Yes indeed, I forgot to add the reference. The reference has been added.
-the definition of tinnitus disorder (De Ridder et al. 2021) and auditory allucinosis (Messina et al. 2022) could be added.
The refence has been added:
(other definitions has been proposed: De Ridder et al., 2021; Messina et al., 2022)
Pag.26: the temporal definition of acute and chronic tinnitus (> 3 months for some authors, > 6 months for other authors) should be added with references and discussed.
References and additional text have been added.
